# Ablative radiotherapy improves survival but does not cure autochthonous mouse models of prostate and colorectal cancer

Daniel R. Schmidt [1,2,3✉], Iva Monique T. Gramatikov [1,2], Allison Sheen[1], Christopher L. Williams[3,4,5], Martina Hurwitz[2,3], Laura E. Dodge [3,6,7], Edward Holupka[2,3], W. S. Kiger III[2,3], Milton R. Cornwall-Brady[1], Wei Huang[1], Howard H. Mak[1], Kathleen S Cormier[1], Charlene Condon[1], K. Dane Wittrup [1], Ömer H. Yilmaz [1,3,8], Mary Ann Stevenson[2,3], Julian D. Down[1], Scott R. Floyd [9], Jatin Roper[1,10] & Matthew G. Vander Heiden [1,4✉]

## Abstract

**Background** Genetically engineered mouse models (GEMMs) of cancer are powerful tools to study mechanisms of disease progression and therapy response, yet little is known about how these models respond to multimodality therapy used in patients. Radiation therapy (RT) is frequently used to treat localized cancers with curative intent, delay progression of oligometastases, and palliate symptoms of metastatic disease. **Methods** Here we report the development, testing, and validation of a platform to immobilize and target tumors in mice with stereotactic ablative RT (SART). Xenograft and autochthonous tumor models were treated with hypofractionated ablative doses of radiotherapy. **Results** We demonstrate that hypofractionated regimens used in clinical practice can be effectively delivered in mouse models. SART alters tumor stroma and the immune environment, improves survival in GEMMs of primary prostate and colorectal cancer, and synergizes with androgen deprivation in prostate cancer. Complete pathologic responses were achieved in xenograft models, but not in GEMMs. **Conclusions** While SART is capable of fully ablating xenografts, it is unable to completely eradicate disease in GEMMs, arguing that resistance to potentially curative therapy can be modeled in GEMMs.

**Plain language summary**

Mice can be used to model the types of cancer seen in people to investigate the effects of cancer therapies, such as radiation. Here, we apply radiation therapy treatments that are able to cure cancer in humans to mice that have cancer of the prostate or colorectum. We show that the mice do not experience many side effects and that the tumours reduce in size, but in some cases show progression after treatment. Our study demonstrates that mice can be used to better understand how human cancers respond to radiation treatment, which can lead to the development of improved treatments and treatment schedules.

[1] Koch Institute for Integrative Cancer Research, Massachusetts Institute of Technology, Cambridge, MA, USA. [2] Department of Radiation Oncology, Beth Israel Deaconess Medical Center, Boston, MA, USA. [3] Harvard Medical School, Boston, MA, USA. [4] Dana-Farber Cancer Institute, Boston, MA, USA. [5] Department of Radiation Oncology, Brigham and Women's Hospital, Boston, MA, USA. [6] Department of Obstetrics and Gynecology, Beth Israel Deaconess Medical Center, Boston, MA, USA. [7] Department of Epidemiology, Harvard T.H. Chan School of Public Health, Boston, MA, USA. [8] Department of Pathology, Massachusetts General Hospital, Boston, USA. [9] Department of Radiation Oncology, Duke University School of Medicine, Durham, NC, USA. [10] Present address: Department of Medicine, Division of Gastroenterology, and Department of Pharmacology and Cancer Biology, Duke University, Durham, NC, USA. ✉email: dschmidt@bidmc.harvard.edu; mvh@mit.edu

Genetically engineered mouse models have enabled the study of many aspects of tumor biology in an endogenous tissue context that are not possible using other approaches[1,2]. In addition to providing a window into the genetic mechanisms leading to cancer development, the ability to probe autochthonous tumors in their native environment enables mechanistic study of how immune cells, stroma, and other factors in the tumor microenvironment contribute to disease progression and treatment response[3,4]. GEMMs are also useful for studying mechanisms of treatment resistance and disease recurrence[5]. Although GEMMs have been used to evaluate treatment efficacy[6–11], they have not yet been extensively used to evaluate response to potentially curative therapies or to examine mechanisms of recurrence post treatment.

Most cancer patients receive multimodality therapy, and understanding how to best sequence existing and novel therapies is an area where GEMMs could impact how clinical trials are designed. It is estimated that approximately 50% of cancer patients receive RT as part of their management, and about half of those patients receive RT with curative intent[12]. Ablative radiotherapy, commonly referred to in clinical practice as SABR (stereotactic ablative body radiotherapy) or SBRT (stereotactic body radiation therapy), is a form of hypofractionated RT in which a small number of fractions are used to deliver definitive doses of radiation to an immobilized and stereotactically localized tumor. Localized tumors can be completely eradicated with SABR/SBRT, allowing cure of early-stage cancers and delayed progression of oligometastatic disease[13]. Clinical factors consistently shown to predict radiation response include tumor size and radiation dose, yet how tumor biology and molecular factors contribute is not well understood. In the SABR-5 oligometastasis trial, local recurrence after SABR occurred in 13% of all cases, but was higher for colorectal cancer oligometastases suggesting that tumor-intrinsic biological factors can contribute to treatment efficacy[14]. Local recurrences also appear to be higher for bone metastases, suggesting tissue environmental factors may also impact response to SABR/SBRT[15]. GEMMs provide an opportunity to study the temporal effects of SABR/SBRT on tumor and host biology and explore how tumor intrinsic and extrinsic factors, including the local immune environment, affect treatment response.

In mice, single fraction or sub-ablative fractionated regimens are well tolerated and can delay tumor growth in preclinical models[16–18]; however, whether ablative doses of radiation are able to cure tumors in GEMMs is not known. In this study we describe the development of a stereotactic RT platform and determine the maximum tolerated dose of hypofractionated RT that can be delivered to the lower abdomen and pelvis in mice. We demonstrate that doses used to fully ablate tumors in the clinic are well tolerated in mice and can result in complete pathologic responses in flank xenografts. In contrast, ablative dose RT is able to reduce tumor growth and improve survival in autochthonous prostate and colorectal tumor models, but does not achieve complete pathologic responses. These data demonstrate that GEMMs can be used to study RT-induced changes in the tumor microenvironment, and suggest that radioresistant persister cells are present in these tumors.

## Methods

**Mouse strains, husbandry, and tumor induction**. All studies were performed according to our animal protocol which was approved by the MIT Institutional Animal Care and Use Committee (Protocol Number 1115-110-18 and 0119-001-22). Mice were housed in a specific pathogen-free (SPF) facility in ventilated microisolator cages at the Koch Institute at MIT with ad libitum access to standard chow and water. Housing rooms maintained a 12-hour light/dark cycle and ambient temperature of 23 °C. C57BL/6 J breeders were purchased from Jackson Laboratories (RRID:IMSR_JAX:000664). CD-1 breeders were purchased from Charles-River (Strain Code: 022). Wild-type mouse colonies were expanded in house and re-crossed with founders from Jackson Laboratories or Charles-River at least once per year to prevent genetic drift. $Pten^{f/f}$ (MGI: 2679886)[19], $Trp53^{f/f}$ (MGI: 1931011)[20], and $Pbsn$-$Cre$ (MGI:2385927)[21] mice were maintained on a mixed C57BL/6 J × 129SvJ background. $Pten^{f/+}$; $Trp53^{f/f}$; $Pbsn$-$Cre$ males were bred with $Pten^{f/f}$; $Trp53^{f/f}$ females to generate $Pten^{f/f}$; $Trp53^{f/f}$; $Pbsn$-$Cre$ male mice for prostate cancer GEMM studies. $Apc^{f/f}$ (MGI: 3688435)[22] and $Villin^{CreERT2}$ (MGI: 3053826)[23] mice were maintained on a pure C57BL/6 J background. To induce colorectal tumors, 50–100 μL of 100 μM 4-hydroxytamoxifen (Calbiochem Cat. # 579002) in PBS was injected into the submucosa of 6–8 week-old male and female $Apc^{f/f}$;$Villin^{CreERT2}$ mice under endoscopic guidance 2–3 cm from the anal verge. Injections were performed using a custom injection needle (Hamilton Inc., 33 G, small Hub RN NDL, 16 inches long, point 4, 45-degree bevel, part # 7803-05), a syringe (Hamilton Inc., part # 7656-01), a transfer needle (Hamilton Inc., part # 7770-02), and a colonoscope with integrated working channel (Richard Wolf 1.9 mm/9.5 French pediatric urethroscope, model # 8626.431). Prostate cancer xenograft studies were conducted with male CD-1 nude mice (Crl:CD1-Foxn1nu/nu) mice purchased from Charles-River laboratories (Strain Code: 086) and acclimatized for at least 2 weeks in the SPF facility at the Koch Institute at MIT. Mice were housed in autoclaved cages with ad lib access to autoclaved chow and water. PC3 cells (RRID:CVCL_0035) were obtained from the Broad Institute cell line repository, STR tested using the ATCC Cell Line Authentication Service, and passaged in DMEM (Corning Cat # 10-013-CV) supplemented with 10% heat-inactivated fetal bovine serum (Sigma). 22Rv1 cells (RRID:CVCL_1045) confirmed by STR testing were kindly donated by Dr. Massimo Loda (Dana Farber Cancer Institute) and passaged in RPMI (Corning Cat # 15-040-SV) supplemented with 2 mM glutamine (Invitrogen) and 10% heat-inactivated fetal bovine serum (Sigma). Both cell lines were routinely tested for mycoplasma. Cells harvested from cultures in exponential growth phase were resuspended in PBS, mixed 1:1 with Matrigel (Corning Cat # 356231) and injected subcutaneously into the caudal flank of 8–10 week-old male CD-1 nude mice. Each flank was injected with 1.5 million PC3 cells or 3 million 22Rv1 cells.

**Radiation delivery and dosimetry**. Radiation was performed on a Gammacell 40 Exactor (Best Theratronics) located in the same SPF facility where the mice were housed. This instrument has a ventilated circular sample chamber (diameter 31.2 cm, height 10.5 cm, volume 8.0 L) that is centered between two Cesium-137 sources located 68 cm apart. During the course of this study the dose rate was between 0.9 and 1.0 Gy per minute. Manufacturer reported dose uniformity for the entire sample chamber is ±7%. Our measurements showed a dose uniformity of ±2.3% in the central zone of the sample chamber where the animal restrainer is placed during treatment. Collimation of the radiation beam reduces the central dose rate by a factor that depends on the size of the collimated field. Based on TLD/OSLD measurements, the output factor for a circular field with a diameter of 1, 2, and 3 cm were 0.62, 0.72, and 0.8 respectively. Independent confirmation of the output factors by radiochromic film dosimetry was within 2% of these estimates. In the shielded zone the dose was attenuated by 95% (±2%).

Mice were immobilized in the restrainer for up to 30 min during radiation delivery. Only one animal was irradiated at a

time. Following irradiation or sham treatment, animals were returned to their home cage with ad lib access to standard chow and water. They were monitored daily for two weeks, and weighed at least once per week for 2 months. Thereafter body condition and activity were assessed once weekly. Any animals showing signs of distress or poor body condition score were promptly euthanized.

Thermoluminescent dosimeters (TLDs) were purchased from Radiation Dosimetry Services at MD Anderson (Houston, TX, USA). TLDs were sandwiched between clinical bolus material (0.5 cm thick), placed in the restrainer in the collimated or open field, and irradiated for 0–6 min. TLDs were promptly returned to Radiation Dosimetry Services for analysis. TLD measurements were performed in triplicate at 3 dose levels. Confirmatory experiments were performed using optically stimulated lumines-cent dosimeters (OSLDs) purchased from Landauer and irra-diated in a similar manner (Supplementary Fig. 1a–c).

Radiochromic film dosimetry was performed using Gafchromic EBT-3 film (Ashland) and Epson 10000XL scanner. Gafchromic EBT-3 film was placed between plexiglass sheets such that the film was located in the central plane of the collimated field (Supplementary Fig. 1d–f). Film dosimetry calibration was generated by exposing film in an open field (between plexiglass sheets placed on a Styrofoam block) in the same irradiator over a dose range between 0 and 8 Gy. The same batch of film and film scanner was used for calibration and measurement, and the orientation of the film and time between exposure and scanning were kept consistent between calibration and measurement. To evaluate the lateral dose distribution, Gafchromic EBT-3 film was placed between plexiglass sheets (6 mm thick) and irradiated with a range of doses between 0 and 8 Gy. In total, 5 films and 6 plexiglass sheets were used to cover the full height of the effective radiation field (Supplementary Fig. 1e). Independent dose-response functions were determined for each of the color channels in the scanned films, and a joint fitting procedure was performed to calibrate the dose profiles. Two independent radiochromic film measurements were performed with 2–3 dose levels per experiment.

**Surgical castration.** Male mice were anesthetized with isoflurane and placed on a warmed surface in the surgical suite. Using aseptic technique, the testis was exposed via 1 cm vertical midline incision in the ventral scrotum and 0.5 cm incision in the tunica. The spermatic blood vessels and vas deferens were cauterized after which the testis and epididymis was removed. Remaining tissue was gently returned into the scrotum and the process was repeated with the other testis via the same scrotal incision. The tunica was closed with tissue glue and the scrotal incision closed with a single wound clip. After recovery from anesthesia mice were returned to a clean cage. Analgesic (Carprofen) was administered once daily for 3 days. Wound clips were removed between day 7–10.

**Radiobiologic calculations.** The linear-quadratic model is used describe the biologic effect of fractionated radiation regimens on tumors and normal tissues[24]. The linear-quadratic (LQ) formula is second-degree polynomial with a linear and a quadratic term that is fitted to empirical clonogenic survival data in order to determine the coefficient of the linear term (alpha) and the coefficient of the quadratic term (beta). In its simplest form, the LQ formula describes the relationship between radiation dose and effect on clonogenic survival

$$radiation\ effect = \alpha D + \beta D^2 \qquad (1)$$

where D is the radiation dose, $\alpha$ is the coefficient of the linear term, and $\beta$ the coefficient of the quadratic term. The ratio of

alpha to beta ($\alpha/\beta$) is the dose at which the linear and quadratic terms contribute equally to the radiation effect, which in turn reflects capacity for DNA repair. Most tumors and proliferating tissues are generally considered to have a high alpha/beta ratio ($\alpha/\beta \geq 10$) while slow growing tumors and quiescent tissues are considered to have a low alpha/beta ratio ($\alpha/\beta \leq 3$). In clinical practice the alpha/beta ratio is used to calculate iso-effective doses for a different dose-fractionation regimen[25]. The formula that is used for this is the biologically effective dose (BED)

$$BED = n \cdot d \left(1 + \frac{d}{\frac{\alpha}{\beta}}\right) \qquad (2)$$

where $d$ is the dose per fraction, $n$ is the number of fractions, and $\alpha/\beta$ is the alpha/beta ratio derived empirically for a given cell line, tumor, or tissue by fitting the survival curve to the LQ formula as described above. In the current study we used $\alpha/\beta = 3$ to calculate the biologically effective dose ($BED_3$) for quiescent tissues where radiation toxicity develops at late time points. For early-responding tissues we used $\alpha/\beta = 10$ to calculate the biologically effective dose ($BED_{10}$). While additional terms can be incorporated in the BED formula to account for time over which the radiation course is delivered and kinetics of tumor repopulation, these are not commonly used in clinical practice[26]. Moreover, the LQ model has been shown to be appropriate for determining iso-effective fractionated radiation regimens with doses up to 18 Gy per fraction, but is less reliable for higher doses and single frac-tion regimens[27].

**Mouse imaging and radiation treatment planning.** Magnetic Resonance Imaging (MRI) was performed on a 7 T MRI system (7 T/210/ASR, Agilent/Varian). Mice were anesthetized by inha-lation of 3% isoflurane and maintained on 1–2% isoflurane throughout data collection with heated air delivery. Axial proton density weighted images were obtained using fast spin echo sequence (fsems) with the following parameters: TR/TE = 2000/ 12 ms, ETL = 4, $256 \times 256$ matrix, FOV = $40 \times 40$ mm$^2$, inter-leaved number of slices = 30–50, no gap and slice thickness = 0.5 mm, number of averages = 2. Scans were collected with respiratory gating (PC-SAM version 6.26 by SA Instruments Inc.) to minimize motion artifact. Contouring of normal tissues for defining the radiation field size and dose-volume analysis was performed in MIM Version 6.0 (MIM Software Inc.).

For image registration and dosimetric studies, an eXplore CT 120 scanner (GE Healthcare) was used to acquire micro computed tomography ($\mu$CT) images with X-ray tube voltage 70kVp, current 50.0 mA, and exposure time of 32 ms. Data was acquired over a 360-degree rotation with a step size of 0.5 degrees. Detector binning of $2 \times 2$ resulted in an isotropic resolution after reconstruction of 50 microns (Parallax Innovations, GPU-based reconstruction). After immobilization in the restrainer, mice were anesthetized by inhalation of 3% isoflurane and maintained on 1–2% isoflurane introduced via nose cone into the front end of the restrainer. Mice were then imaged by MRI, followed immediately by CT. Contours of abdominopelvic organs were generated on axial MRI images. MRI-CT image registration was performed in MIM. MRI and CT images were co-registered using rigid fusion to the upper pins of the restrainer, after which the MRI structure set was transferred to CT. For pelvic radiation, the treatment isocenter was placed mid-plane between the top and bottom of the restrainer and centered 3 mm above the top edge of the upper pins for the 1 cm circular field, or 8 mm above the top edge of the upper pins for the 2 cm circular field. A Monte Carlo dose distribution model based on radiochromic film dosimetry was generated. In brief, we modeled the Gammacell irradiator and our collimation system in the EGSnrc software

package[28]. Hounsfield units for the mouse CT scanner were calibrated annually to air, water, and bone-like material using a phantom provided by the manufacturer. CT Hounsfield Units were used to classify the tissue type in each voxel, and to scale the physical density in each voxel based on a generic CT density curve for the purposes of Monte Carlo dose calculation. Monte Carlo dose distributions were normalized using the OSLD dosimetry measurements, and validated through comparison with beam profiles extracted from our film dosimetry. Treatment plans for a parallel opposed beam arrangement were generated for male and female mice and both 1 cm and 2 cm diameter circular radiation fields. Dose (37.5 Gy) was prescribed to the isocenter. Dose-volume histograms were generated for all major abdominopelvic organs, and the maximum, minimum, and mean doses for each organ were calculated. MIM was used for treatment planning and generating dose-volume histograms. VelocityAI version 3.1 (Varian Medical Systems Inc.) was used for visualization of the dose distribution and MRI-defined structures.

For assessment of inter-fraction setup error, conscious animals were secured in the restrainer and imaged on the eXplore CT 120 µCT scanner. Repeat scans for the same animal were obtained on 3 separate days. DICOM files were imported into MIM and bony anatomy was outlined on each scan using the threshold tool. Images were co-registered using rigid fusion to the upper pins of the restrainer. Contours of the bony anatomy were transferred to a single CT scan to compare alignment of the pelvic bones between scans.

For assessment of intra-fraction motion, a Skyscan 1276 (Bruker) was used to acquire serial images with a stationary gantry. In total, 1200 images were acquired over a period of approximately 3 min. Images were acquired with x-ray tube settings of 100 kV, 200 mA, and an exposure time of 33 ms with a 0.5 mm aluminum beam filter. $8 \times 8$ detector binning was used for an isotropic resolution after reconstruction of 80.3 microns in NRecon 2.0 (Bruker). Mice were imaged twice in succession; first while awake and free breathing, then immediately thereafter while under 2.5% isoflurane anesthesia. The two-dimensional (2D) grayscale images collected were analyzed to determine the amount of movement during each imaging period. It was observed that the subjects would experience brief periods of movement separated by relatively long motionless intervals. Calculating a median value for each pixel position during the imaging period results in a neutral position representing the neutral, resting position of the subject. With this as a reference, the deviation from the neutral position was calculated for each image by first applying a threshold for bone to both images and then calculating the absolute value of the difference of the two images. The thickness of non-zero regions of this image were taken to represent the distance the bone moved. The maximum thickness, representing maximum displacement, in each image was used to represent the overall movement in that image. This was expressed as a percent time spent at a given displacement from the neutral position. Further, this number was limited to include only a 2 cm diameter region centered in the pelvis. Data analysis was done in MATLAB (MathWorks).

For bioluminescence imaging, mice were injected subcutaneously with 150 mg/kg D-luciferin (Caliper Life Sciences). After 10 min they were anesthetized by inhalation of 3% isoflurane and maintained on 1.5% isoflurane introduced via nose cone. Mice were imaged on an IVIS Spectrum in vivo optical imaging system (PerkinElmer) equipped with a heated platform to maintain a physiological body temperature during imaging. Images were collected every 5 min over a 40-min period in order to capture the period of maximum photon emission. Living Image 3.1 (PerkinElmer) was used for image processing.

For PET imaging, mice were fasted overnight and dosed intravenously with ~80 µCi of F-18 fluorodeoxyglucose (PETNET Solutions, Woburn, MA) via tail vein. Mice were kept anesthetized at 2% isoflurane for 1 h uptake at 37 °C and then imaged for 10 min PET acquisition and 2 min CT using a G8 PET/CT system (Perkin Elmer). Images were reconstructed using default MLEM 3D protocol and CT attenuation correction and were visualized in AMIDE Version 1.0.5.

**Colonoscopy and volumetric analysis of tumors**. Optical colonoscopy was used to document colorectal mucosal toxicity in the dose escalation studies and to monitor tumor size in the colorectal adenoma studies, as previously described[29,30]. Endoscopic evaluation of tumor size was performed three weeks after tumor induction. Animals were stratified into 4 groups by baseline tumor size and the largest and smallest groups were excluded. Animals in the two middle groups were separately randomized to radiation or sham treatment to ensure similar baseline tumor size between groups. Tumors were imaged with optical colonoscopy before radiation treatment and after. Offline images were analyzed with FIJI[31] and the Tumor Size Index (TSI) was calculated as

$$\frac{tumor\ area}{lumen\ area} \times 100 \qquad (3)$$

For xenograft studies the length, width, and height of xenografts was measured using digital calipers. All measurements were performed by a single operator. Tumor size was calculated as the volume of an ellipsoid

$$tumor\ volume = \frac{4}{3}\pi \cdot \left(length/2\right) \cdot \left(width/2\right) \cdot \left(height/2\right) \quad (4)$$

MRI was used for autochthonous prostate tumor monitoring. A coronal scout was obtained to estimate tumor size after which axial images were collected one slice inferior to the pubic symphysis to one slice above the cranial edge of the tumor. Tumors were contoured using the ROI tool in OsiriX Version 7.0.2 (Pixmeo Sarl) to determine tumor volume.

**Histology and Immunohistochemistry**. Tissues collected at time of necropsy were fixed in 10% neutral buffered formalin for a minimum of 3 days at 4 °C. Formalin fixed tissues were dehydrated, paraffin embedded, sectioned, and stained with hematoxylin and eosin (H&E) or Masson's Trichrome. For whole-mount histology, a male mouse was removed from the restrainer after CO2 euthanasia, fixed in Bouin's fixative for 1 week at 4 °C, decalcified in 0.5 M EDTA pH 8.0 solution for 3 weeks, step sectioned (approximately 750 µm between sections) and H&E stained.

For IHC the following primary antibodies were used: Ki67 (Biocare Medical, Cat# CRM 325, RRID:AB_2721189) diluted 1:50 in TBST, Cleaved Caspase-3 (Asp175) (Cell Signaling, Cat# 9664, RRID:AB_2070042) diluted 1:800 in TBST, CD8 (Cell Signaling, Cat# 98941, RRID:AB_2756376) diluted 1:400 in TBST, FoxP3 (Cell Signaling, Cat# 12653, RRID:AB_2797979) diluted 1:200 in TBST, Phospho-Histone H2A.X (Ser139) (Cell Signaling, Cat# 9718, RRID:AB_2118009) diluted 1:600 in TBST, p21 (Abcam, Cat# ab188224, RRID:AB_2734729) diluted 1:1000 in TBST. For γH2AX and p21 IHC 5 µm thick formalin-fixed paraffin-embedded (FFPE) sections were deparaffinized, rehydrated, and immediately underwent heat-mediated antigen retrieval a pressure cooker (Biocare Medical) at 125 °C for 5 min in Citra pH 6.0 solution (Biogenex, HK086) for γH2AX, or in Tris-EDTA (pH 9.0) antigen retrieval solution (Abcam, Cat # ab93684) for p21. After cooling to room temperature, sections were equilibrated in distilled water prior to processing.

Endogenous peroxidase activity was quenched with BLOXALL (Vector Labs, SP-6000) for 20 min. Sections were then blocked for 30 min with 3 percent normal goat serum, incubated overnight with primary antibody at 4 °C, incubated with avidin/biotin/HRP reagents per manufacturer recommended protocol (Vector Labs, ABC-HRP Kit, Cat# PK-4001), incubated with DAB substrate (Vector Labs, Cat# SK-4100) for 5 min at room temperature, and counterstained with hematoxylin. Ki67, Cleaved caspase-3, CD8, and FoxP3 IHC was performed on the Thermo Scientific LabVision 360 autostainer. Antigen retrieval was done using the PT module (LabVision) at 97 °C for 20 min in citrate buffer pH 6 (Abcam, Cat# 3678), followed UltraVision Hydrogen Peroxide Block (ThermoScientific, TA-125-H202), Rodent Block M (Biocare Medical, #RBM961), primary antibody for 1 h, Rabbit on Rodent HRP (Biocare Medical #RMR622), DAB Quanto (ThermoScientific, #TA-125-QHDX) for 5 min, and counterstained with hematoxylin. For visualization and image processing, slides were scanned using an Aperio AT2 digital slide scanner (Leica Biosystems) at 20X magnification.

For γH2AX immunofluorescence, primary antibody incubation was done as above. Sections were then incubated in the dark at room temperature for 30 min with goat anti-rabbit Alexa Fluor 568 (Invitrogen Cat # A-11036, RRID:AB_10563566) diluted 1:50 in TBST, mounted in VECTASHIELD with DAPI (Vector Laboratories, Cat # H-1200), and imaged with an inverted fluorescence microscope (Nikon Eclipse Ti-S, SPOT RT3 camera, SPOT5.2 software). Images were taken with 10X objective in phase contrast, DAPI, and TRITC channels. Channels were merged in ImageJ Version 1.53a (NIH).

Senescence-associated β-galactosidase staining was performed according to previously published protocol[32]. Freshly collected tissue was immersed in OCT and frozen on dry ice. A cryotome was used to cut 4 μm thick sections that were fixed in 1% paraformaldehyde in PBS for 1 min at room temperature, rinsed in PBS, and incubated in a CO2 free incubator at 37 °C for 12–16 h in X-gal staining solution, which was prepared fresh as described[32]. Thereafter sections were counterstained with eosin.

Percent Ki67 positive cells in adenomas was quantitated using an automated algorithm in QuPath Version 0.3.0[33]. The analyzer was blinded to treatment group. Three high power fields per adenoma (image perimeter 2.5 mm) were selected from the most cellular portion with highest gland to stroma ratio. Total number of tumor cells and Ki67 positive cells per high power field were quantitated using the analyze -> cell detection -> positive cell detection function with the following parameters: setup parameters (optical density sum, pixel size 0.5 μm), nucleus parameters (background radius 8 μm, median filter radius 0 μm, sigma 1.5 μm, min area 10 μm2, max area 400 μm2), intensity parameters (threshold 0.1, max background intensity 2, split by shape selected), cell parameters (cell expansion 5 μm, include cell nucleus selected), general parameters (smooth boundaries selected, make measurements selected), intensity threshold parameters (nucleus DAB OD mean, single threshold). Percent p21 positive cells in adenomas was quantitated in QuPath as described above with the following modifications. Three high power fields (image perimeter 2 mm) per adenoma were selected from the luminal region of the adenoma and three high power fields from the region adjacent to the muscularis propria (base). CD8+ and FOXP3+ cells per unit area were quantitated in QuPath as described above with the following modifications. Three high power fields per adenoma (image perimeter 2 mm) were selected from the most cellular portion with highest gland to stroma ratio. The same areas were used to quantitate CD8+ and FOXP3+ cells using the analyze -> cell detection -> positive cell detection function. For CD8 the score compartment was Cell DAB OD mean. For FOXP3 the score compartment was Nucleus DAB OD mean. A second analysis was performed to separately quantify CD8+ cells in intraepithelial/tumoral and stromal compartments. Six high power fields were analyzed with manual assignment of CD8+ cells to intraepithelial/tumoral or stromal compartments. This was not done for FOXP3+ cells since they were exclusively located in the stromal compartment.

Ki67+, CD8+, and FOXP3+ cells in prostate tumors were quantitated using the same protocol as described above for adenomas. The analyzer was blinded to treatment group. Ten high power fields per tumor (image perimeter 2 mm) were selected at random throughout the tumor, avoiding areas of necrosis and areas containing normal glands. In addition, a separate analysis was performed for CD8+ cells in peripheral regions of the tumor near to normal glands (5 high power fields per tumor).

**Hematology and serum chemistry**. Whole blood was collected as a terminal procedure. Mice were euthanized by CO2 inhalation. After confirming absence of respiration, venous blood was collected from the inferior vena cava using a 26 G needle and 200 μL immediately transferred to an EDTA tube (Sarstedt, Item # 20.1288.100). Blood counts and hematology profiles were measured in the Division of Comparative Medicine at MIT on a HemaVet 950 FS (Drew Scientific) within a few hours of blood collection. To measure serum testosterone, whole blood collected as above was transferred to a serum separator tube (BD Biosciences, Cat # 365967), allowed to clot for 30 min at room temperature, and centrifuged for 90 s at $10,000 \times g$. Serum was stored at −80 °C until day of assay. Serum testosterone was measured by ELISA in 96-well plate format according to manufacturer recommendations (Cayman, Item # 582701). Absorbance (410 nm) was measured at 90 min on a Tecan infinite 200Pro. For standards, absorbance was plotted versus testosterone concentration and the curve was fitted by a 4-parameter logistic model, which was used to determine the testosterone concentration in serum samples. Calculations were performed in Prism version 8 (GraphPad Software).

**Flow cytometry**. Cells were stained using antibodies to CD4 (BD Biosciences Cat# 612761, RRID:AB_2870092), CD19 (BD Biosciences Cat# 550992, RRID:AB_398483), CD45 (BioLegend Cat# 103116, RRID:AB_312981), CD3 (BioLegend Cat# 100220, RRID:AB_1732057), CD8a (BioLegend Cat# 100759, RRID:AB_2563510), FOXP3 (BioLegend Cat# 126404, RRID:AB_1089117), CD25 (BioLegend Cat# 102012, RRID:AB_312861), CD19 (BioLegend Cat# 152409, RRID:AB_2629838), and NK1.1 (BioLegend Cat# 108705, RRID:AB_313392). All antibodies were diluted 1:100. Viability was assessed using Zombie Aqua (Biolegend, Cat# 423101) or Zombie UV (Biolegend, Cat# 423107) diluted 1:1000. Intracellular staining for FoxP3 was performed using the eBioscience FoxP3 Transcription Factor Buffer Set (ThermoFisher, Cat# 00-5523-00).

Blood, spleen, pelvic and axillary lymph nodes were harvested 2 days or 8 weeks after the last dose of radiation. All tissue samples were weighed and kept in RPMI media (ATCC, Cat# 30-2001) on ice during collection. Spleen and lymph nodes were mechanically digested through 70 um nylon cell strainers to prepare single-cell suspensions for staining. Red blood cells in spleen and blood samples were lysed in ACK Lysing Buffer (Gibco, Cat# A10492-01). All samples were resuspended in ice-cold PBS and stained for viability, then resuspended in ice-cold PBS containing 1% (w/v) BSA and 2 mM EDTA before labeling. Cells were analyzed using BD FACS LSR Fortessa (BD Biosciences), or BD FACS Symphony A3 (BD Biosciences) flow

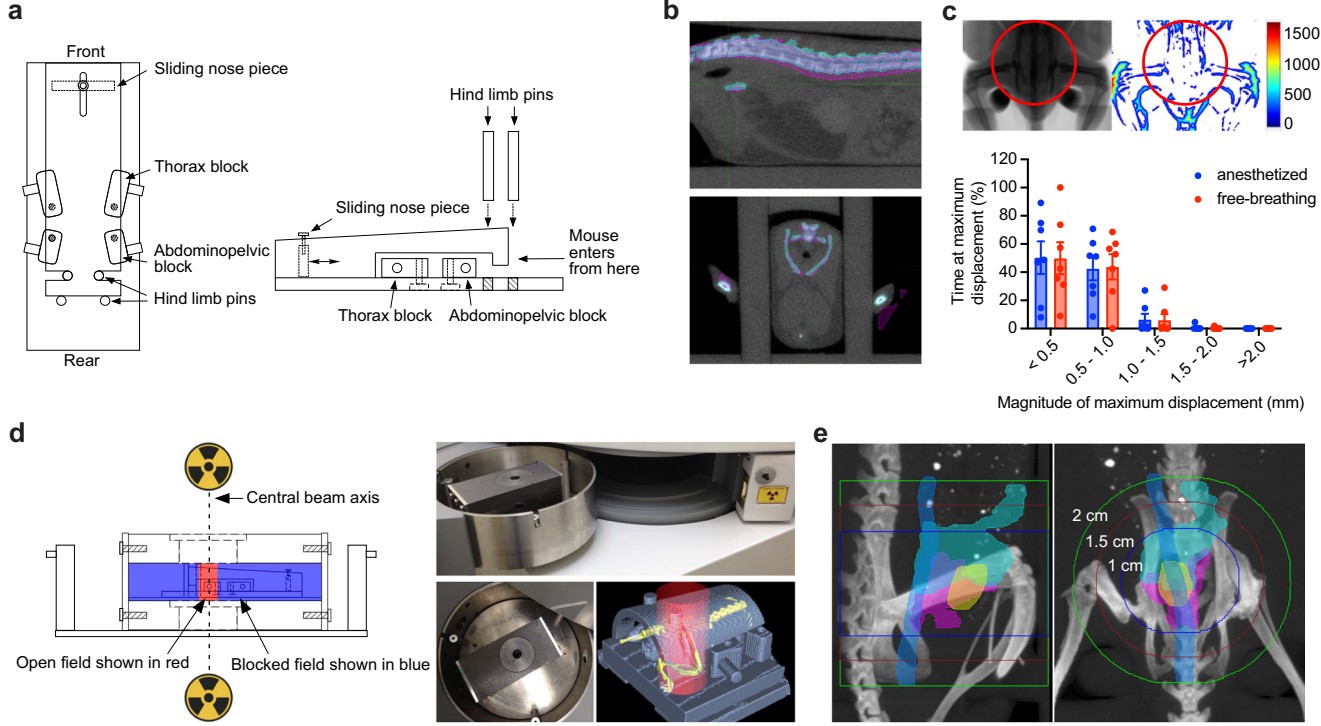

**Fig. 1 Developing a stereotactic radiotherapy (RT) platform. a** Schematic of restrainer (top and side view) for delivery of stereotactic RT. **b** Serial imaging demonstrates negligible anatomic displacement between fractions. The pink and cyan colors indicate the bony anatomy of a mouse imaged on two separate days in the restrainer, with overlap demonstrated for the spine on sagittal view (upper panel) and pelvis on axial view (lower panel). **c** Intrafraction motion assessed by fluoroscopic imaging. The representative anatomic heatmap on the right shows the maximum displacement (in µm) of a free-breathing mouse during 3 min of continuous imaging. The histogram shows the percent of time that a maximum displacement occurs in the pelvic radiation field (area within the red circle). Data are plotted as mean with standard deviation ($n = 7$ per group). **d** Schematic and photographs of lead shields with interchangeable collimators (2 cm aperture shown) used to focus radiation sources located above and below the animal. Example of a focal radiation field (red cylinder) targeted to the pelvis of an immobilized mouse (skeleton in yellow). **e** Visualization of field size relative to pelvic organs of a male mouse. Organs were outlined on MRI and overlaid with bone windows from CT after rigid image registration. Organs depicted include bladder (yellow), prostate (pink), seminal vesicles (cyan), and colorectum (blue).

cytometers. Data was analyzed in FlowJo version 10 (BD Biosciences). A representative example of the gating strategy is shown (Supplementary Fig. 2a).

**Statistics and reproducibility**. Continuous data were first assessed using the D'Agostino-Pearson normality test. Data that passed the normality test were then evaluated using two-tailed Student's *t* test. Independent data utilized unpaired tests, while dependent data utilized a paired test. Data that were not normally distributed were evaluated using the Mann–Whitney U test. Survival curves were compared using a log-rank test. The sample sizes, number of replicates, statistical test used, and the level of statistical significance applied are described in each figure legend. In most figures, only *P* values < 0.05 are noted. Unless otherwise indicated, data are presented as mean ± standard deviation. Prism Version 9 (GraphPad Software) was used for statistical analyses and data visualization.

**Reporting summary**. Further information on research design is available in the Nature Portfolio Reporting Summary linked to this article.

## Results
**Design of a stereotactic radiotherapy platform**. Since autochthonous tumors in GEMMs generally develop in a defined anatomic location and progress at a characteristic rate, we reasoned that a method for target immobilization would allow

accurate delivery of multi-fraction RT regimens without anesthesia or daily image guidance. To this end we designed a series of modular restrainers to allow accurate stereotactic positioning of laboratory mice within the treatment field, minimize target movement during treatment, and accommodate mice of various sizes (Fig. 1a, Supplementary Fig. 3a, Supplementary Methods). We found that the restrainers enabled consistent anatomic alignment in the treatment field (Fig. 1b), minimal intrafraction movement in a conscious animal (Fig. 1c, Supplementary Movie 1), and reproducible positioning of male and female mice spanning a wide range of body weights (Supplementary Fig. 3b). Importantly, animals tolerated immobilization in the restrainer for up to 30 min without overt signs of distress or changes in body weight. An accompanying set of lead shields and circular collimators with apertures ranging from 0.5 to 6 cm were designed to allow focal radiation of the immobilized target via a radiation source located above and/or below the animal (Fig. 1d, Supplementary Methods).

We next determined whether landmarks on the restrainer could be used to accurately localize the anatomic region to be targeted with RT. Autochthonous tumors arising in the prostate of genetically engineered mouse models can occur simultaneously in all prostate lobes. While the anterior and ventral lobes are easily identified on magnetic resonance imaging (MRI), the lateral and dorsal lobes are less apparent. In order to define the borders of the prostate and its relationship to bony anatomy, both computed tomography (CT) and MRI were correlated with whole

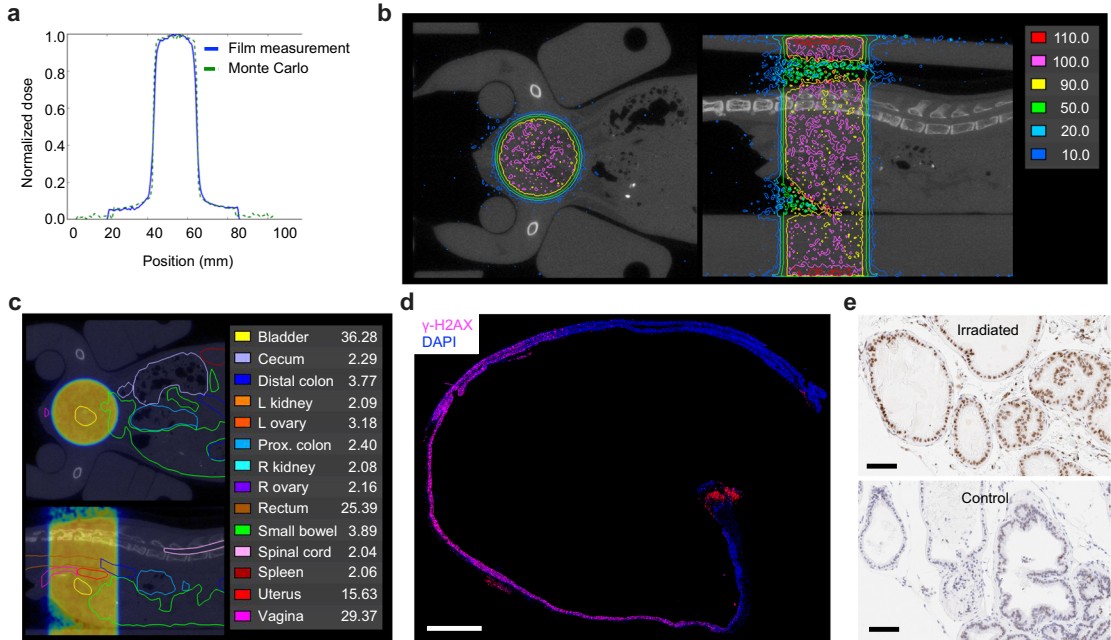

**Fig. 2 Modeling and verification of radiation dose distribution in mice. a** Comparison of measured and modeled dose distribution for a 2 cm radiation field. **b** Representative dose distribution of a 1 cm circular pelvic radiation field mapped onto coronal (top view) and sagittal (side view) CT image of a female mouse. The reference isodose (100%) is shown in pink. Areas enclosed by the red line receive a dose >110% of the reference dose. Areas outside the dark blue line receive a dose <10% of the reference dose. **c** Coronal and sagittal abdominopelvic CT image of a female mouse with organs outlined as indicated. Mean organ dose in Gy is shown for a prescribed dose of 37.5 Gy. Dose distribution of a 1 cm circular pelvic radiation field is shown as a dose wash to allow visualization of organs. Dose color scheme as in (**b**). **d** Evaluation of focal DNA damage in a longitudinal cross-section of colorectum by γ-H2AX staining. This animal was treated with 15 Gy to a 2 cm field centered in the pelvis 30 min prior to tissue collection. Nuclear γ-H2AX foci are seen only in the portion of the colorectum that was within the radiation field. The red signal at the anus is due to autofluorescence. Composite image was generated by stitching 24 individual images taken with a 10X objective. Scale bar = 2 mm. **e** Evaluation of focal DNA damage by γ-H2AX IHC in prostate glands. The animal was treated with 15 Gy to a 2 cm field centered over the prostate 30 min prior to tissue collection. The lower panel shows prostate tissue from an unirradiated control. Scale bar = 100 μm.

mount histologic sections (Supplementary Fig. 4a). These studies showed that the inferior border of the prostate extends to approximately 2–3 mm above the pubic symphysis and the geometric center of the prostate is approximately 2–3 mm above the upper edge of the acetabulum (Fig. 1e, Supplementary Fig. 4b). Since both of these bony landmarks can be reproducibly positioned relative to reference points on the restrainer, we used them as a surrogate for the anatomical position of the prostate. We next identified the optimal position of the restrainer relative to the collimator aperture in order to center the prostate in the radiation field (Supplementary Fig. 4c). Using reference points on the restrainer, the center of the prostate was able to be reproducibly aligned in the center of the radiation field (Supplementary Fig. 4d). Although a 1.5 cm diameter field is sufficient to target the entire normal prostate gland with a 2–3 mm margin, the size of the prostate increases with tumor growth and therefore a 2 cm diameter field was used to target autochthonous prostate tumors of early to intermediate stage (Fig. 1e, Supplementary Fig. 4b).

**Dosimetry and verification of dose distribution in vivo.** For the RT platform to be generalizable, we used a widely available gamma-ray irradiator in which the irradiation chamber is centered between two motorized Cesium-137 sources located above and below the chamber (Fig. 1d). An advantage of this geometry is that the dose distribution within the animal is more homogeneous compared to a single beam arrangement in which the entrance dose can be significantly higher than the deep tissue dose due to beam attenuation. In addition, the narrow energy distribution of Cesium-137 radiation with a peak at 662 keV

allows for a relative reduction in skin dose in the dose buildup region (~1 mm), thus reducing dermatologic toxicity. A combination of optically stimulated luminescent dosimeters (OSLDs) and radiochromic film dosimetry were used to assess absolute dose and the dose distribution of the collimated radiation field (Supplementary Fig. 1a–f). Within the collimated field, radiochromic film dosimetry showed a homogeneous dose distribution and narrow penumbra (Fig. 2a and Supplementary Fig. 5a–c).

MRI is optimal for anatomic evaluation of the mouse abdomen and pelvis; however, it lacks electron density information needed to calculate absorbed radiation dose. We therefore used both MRI and CT imaging to generate an accurate spatial dosimetric analysis (dose-volume relationship) for abdominopelvic organs. Organs outlined on MRI were registered to CT to allow for dose calculation (Fig. 2b, c and Supplementary Fig. 5d). Treatment plans were generated for male and female mice with 1 and 2 cm radiation fields centered in the pelvis to simulate treatment of the prostate and rectum. Critical organs such as the kidneys and spinal cord received <10% of the prescribed dose (Fig. 2c and Supplementary Fig. 5e–h). Phosphorylation of histone H2A.X at Ser139 (γ-H2AX) is a sensitive marker of DNA double strand breaks induced by ionizing radiation[34], and γ-H2AX immunohistochemistry (IHC) can be used to evaluate the focal effects of radiation in tissue. Consistent with the dosimetric predictions, we found homogeneous induction of γ-H2AX in prostate tissue and the portion of the colorectum that was within the radiation field, but not in adjacent colon and distal rectum that were outside the radiation field (Fig. 2d, e). These data demonstrate that with appropriate collimation and immobilization, stereotactic focal radiation can be delivered to mouse models using

**Table 1 Summary of dose escalation studies.**

| Dose level | Number of fractions | Dose per fraction (Gy) | Total Dose (Gy) | BED α/β = 3 | BED α/β = 10 | Elapsed days | Days of treatment | Field size | Strain | Incidence of severe toxicity/death | | Death post SART (weeks) |
|---|---|---|---|---|---|---|---|---|---|---|---|---|
| | | | | | | | | | | Male | Female | |
| DL1 | 1 | 15 | 15 | 90 | 38 | N/A | 1 | 2 cm | C57Bl/6 | 0 of 3 | | |
| | 2 | 10.2 | 20.4 | 90 | 41 | 8 | 1,8 | 2 cm | C57Bl/6 | 0 of 3 | | |
| | 3 | 8.1 | 24.3 | 90 | 44 | 9 | 1,5,9 | 2 cm | C57Bl/6 | 0 of 3 | | |
| | 5 | 6 | 30 | 90 | 48 | 9 | 1,3,5,7,9 | 2 cm | C57Bl/6 | 0 of 3 | 0 of 3 | |
| DL2 | 1 | 18.3 | 18.3 | 130 | 52 | N/A | 1 | 2 cm | C57Bl/6 | 0 of 3 | | |
| | 2 | 12.6 | 25.2 | 131 | 57 | 8 | 1,8 | 2 cm | C57Bl/6 | 0 of 3 | | |
| | 3 | 10 | 30 | 130 | 60 | 9 | 1,5,9 | 2 cm | C57Bl/6 | 0 of 3 | | |
| | 5 | 7.5 | 37.5 | 131 | 66 | 9 | 1,3,5,7,9 | 2 cm | C57Bl/6 | 0 of 3 | 1 of 3 | 16 |
| | 5 | 7.5 | 37.5 | 131 | 66 | 5 | 1,2,3,4,5 | 2 cm | C57Bl/6 | 1 of 5 | 3 of 5 | 12–16 |
| | 5 | 7.5 | 37.5 | 131 | 66 | 9 | 1,3,5,7,9 | 2 cm | CD-1 | 0 of 6 | | |
| | 5 | 7.5 | 37.5 | 131 | 66 | 9 | 1,3,5,7,9 | 1 cm (pelvis) | CD-1 | 0 of 6 | | |
| | 5 | 7.5 | 37.5 | 131 | 66 | 9 | 1,3,5,7,9 | 1 cm (low abd) | C57Bl/6 | | 1 of 4 | 14 |
| DL3 | 1 | 21.8 | 21.8 | 180 | 69 | N/A | 1 | 2 cm | C57Bl/6 | 0 of 2 | | |
| | 2 | 15 | 30 | 180 | 75 | 8 | 1,8 | 2 cm | C57Bl/6 | 0 of 3 | | |
| | 3 | 12 | 36 | 180 | 79 | 9 | 1,5,9 | 2 cm | C57Bl/6 | 0 of 3 | | |
| | 5 | 9 | 45 | 180 | 86 | 9 | 1,3,5,7,9 | 2 cm | C57Bl/6 | 1 of 3 | 2 of 3 | 7–12 |
| | 5 | 9 | 45 | 180 | 86 | 9 | 1,3,5,7,9 | 1 cm (pelvis) | C57Bl/6 | | 0 of 4 | |
| | 5 | 9 | 45 | 180 | 86 | 9 | 1,3,5,7,9 | 1 cm (low abd) | C57Bl/6 | | 2 of 4 | 9–10 |
| DL4 | 2 | 18 | 36 | 252 | 101 | 8 | 1,8 | 2 cm | C57Bl/6 | 0 of 3 | | |
| | 3 | 14.4 | 43.2 | 251 | 105 | 9 | 1,5,9 | 2 cm | C57Bl/6 | 3 of 3 | | 14–17 |
| | 5 | 10.9 | 54.5 | 253 | 114 | 9 | 1,3,5,7,9 | 2 cm | C57Bl/6 | 3 of 3 | | 5 |

Dose levels and fractionation schemes evaluated in the pelvic stereotactic ablative radiotherapy (SART) dose escalation studies are shown, including number of fractions, dose per fraction, cumulative nominal dose, BED, days on which radiation was delivered, diameter of the circular radiation field, mouse strain, and incidence/timing of severe toxicity or death occurring within 6 months of treatment. Some cohorts were monitored longer than 6 months. No treatment-related deaths occurred in these cohorts. Elapsed days is the total number of days over which radiation was delivered, including the first and last day. Radiation fractions were equally distributed across the elapsed days. The single fraction regimen at DL4 was not evaluated because the length of time in the restrainer would have exceeded the maximum time approved under our animal protocol. The 2 cm radiation field was centered in the pelvis over the prostate and/or bladder (see "Methods" for details). The 1 cm radiation field covered either the lower half (pelvis) or upper half (low abdomen) of the 2 cm field. BED and α/β are further described in the "Methods".
*DL* dose level, *α/β* alpha-beta ratio, *BED* biologic effective dose, *N/A* not applicable.

a variety of radiation sources, including widely available gamma-ray irradiators.

**Determining the maximum tolerated dose of hypofractionated pelvic SART**. To evaluate the tolerance of pelvic organs in mice to hypofractionated SART, we performed a series of dose escalation studies (summarized in Table 1). Single fraction and 2, 3, and 5-fraction regimens were evaluated at 4 dose levels for males and 5-fraction regimens at 3 dose levels for females. The biologically effective dose (BED) model[25] was used to calculate isoeffective doses for the different fractionation schemes (see methods for details). Isoeffective dose levels were set to account for late effects of radiation that can cause permanent damage to normal tissues leading to organ failure (Supplementary Fig. 6a). Within each dose level, regimens with higher number of fractions, and thus higher total dose, are anticipated to have greater effects on proliferating tumor cells, as well as proliferating cells in normal tissues, leading to early-onset toxicity in bone marrow, intestine, and skin (Supplementary Fig. 6a). One limitation of the BED model is that it does not account for time effects, yet it is well established that accelerated radiation regimens are associated with greater incidence of acute toxicity. Indeed, we observed increased acute to subacute gastrointestinal toxicity when a 5-fraction regimen was delivered daily rather than every other day (QOD) resulting in higher incidence of treatment-related mortality (Table 1). In order to limit toxicity and to control for time effects across regimens, all fractionated regimens were delivered over 8-9 days with equal number of days between each fraction (see Table 1).

To determine the maximum tolerated dose of hypofractionated pelvic RT, 3 mice were treated at a given dose level (DL) and were monitored for a minimum of 6 months prior to moving to a higher dose level. The maximum tolerated dose was defined as the highest dose at which at least 2 of 3 animals survived, which in males was found to be 9 Gy x 5 fractions and in females was 7.5 Gy x 5 fractions (Table 1 and Fig. 3a). In line with predictions from the BED calculations, we found that acute gastrointestinal and dermatologic toxicity occurred more frequently, and with greater severity, in animals treated with higher numbers of fractions. All animals that received 10.9 Gy x 5 fractions showed signs of acute dermatologic toxicity (alopecia and desquamation) that peaked between 2–3 weeks after radiation (Fig. 3b). At this dose, acute gastrointestinal toxicity was also apparent, including colorectal edema (Fig. 3c), loose stools, and hematochezia, which was accompanied by persistent weight loss that resulted in death or significant decrease in body condition requiring euthanasia within 5 weeks of completing treatment. (Fig. 3a and Supplementary Fig. 6b). Necropsy revealed focal areas of bleeding and ulceration in the distal colon and rectum that was grossly apparent and confirmed by histology (Fig. 3d and Supplementary Fig. 6c). The bladder showed signs of mild-moderate edema (submucosal thickening) but no loss of epithelial integrity (Supplementary Fig. 6d). The prostate and seminal vesicles appeared largely normal, but in some cases had reduced luminal secretions (Supplementary Fig. 6d), while the testis showed impaired spermatogenesis (Supplementary Fig. 6d). Animals treated with the equivalent 3-fraction regimen (14.4 Gy x 3 fractions, Table 1 DL4) died or required euthanasia within 14–17 weeks after radiation (Table 1 and Supplementary Fig. 6b). Necropsy revealed histologic evidence of chronic damage to the colorectum (Fig. 3e). Animals treated with the equivalent 2-fraction regimen (18 Gy x 2 fractions, Table 1 DL4) showed no signs of acute toxicity other than transient weight loss;

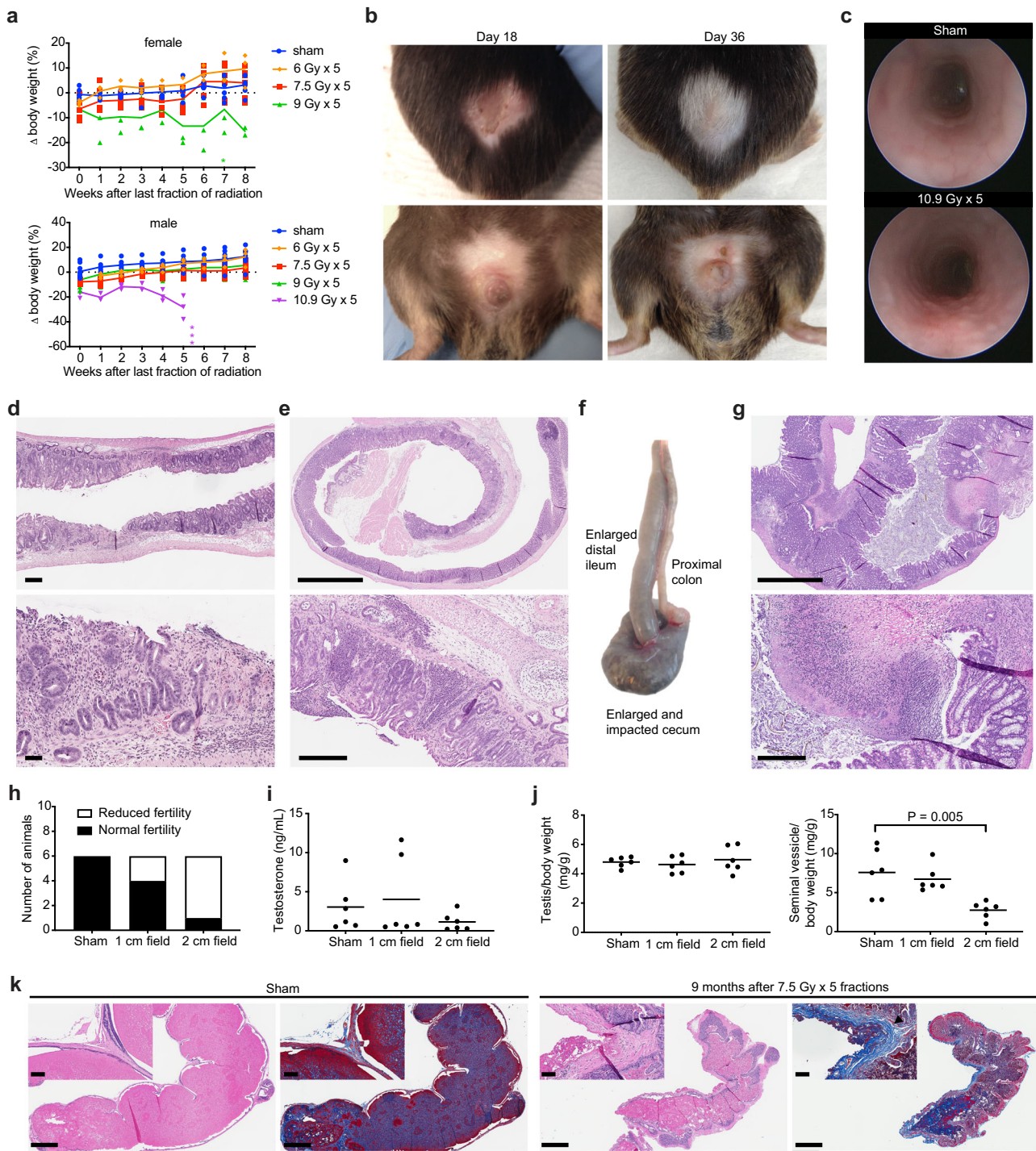

however, there was evidence of chronic tissue damage at planned necropsy 6 months after treatment, including focal atrophy of prostate and seminal vesicles, fat necrosis in the epididymal fat pad, and focal atrophy of seminiferous tubules in one testis in 1 of 3 animals. (Supplementary Fig. 6d–f).

In contrast to the overt signs of acute gastrointestinal and dermatologic toxicity observed at the highest dose level tested, the only consistent sign of acute toxicity at lower dose levels was decreased activity and characteristic weight loss that peaked two days after completing radiation (Supplementary Fig. 7a). Interestingly, of the animals presenting with toxicity at the second highest dose level tested (Table 1, DL3), there was no evidence of proctitis, but rather chronic inflammation and loss of epithelial

integrity in the cecum, and in some cases concurrent enlargement of the distal ileum (Fig. 3f, g and Supplementary Fig. 7b). Overall, acute and chronic gastrointestinal toxicity occurred more frequently in female mice, which could be mitigated by reducing the volume of cecum and small bowel in the treatment field (Supplementary Fig. 7c, d).

For animals in the lowest dose level tested (Table 1, DL1), the late effects of radiation were assessed at 12 months. Animals in the intermediate dose levels (Table 1, DL2 and DL3) were assessed at 6 months. Some animals at DL2 and DL3 showed small foci of fat necrosis or fibrosis, but there were no other obvious signs of chronic tissue damage. Histologic evaluation showed no signs of chronic inflammation or fibrosis of the

**Fig. 3 Acute dose-limiting toxicity and late effects of hypofractionated stereotactic ablative radiotherapy (SART) to the pelvis. a** Body weight change relative to pretreatment baseline for female and male mice treated with 6–10.9 Gy x 5 fractions QOD. Individual animals are shown with group means connected. Female $n = 6$ for 0 and 7.5 Gy, $n = 3$ for 6 and 9 Gy. Male $n = 9$ for 0, 7.5, and 9 Gy, $n = 3$ for 6 and 10.9 Gy. Each asterisk (*) indicates time of death or protocol mandated euthanasia of a single animal. **b** Representative skin changes in a male mouse at the indicated time points after completing 10.9 Gy x 5 fractions QOD to a 2 cm pelvic field. Upper panels show dorsal side, lower panels ventral side. **c** Endoscopic images show colorectal mucosa in an untreated animal and 1 week after completing 10.9 Gy x 5 fractions QOD to a 2 cm pelvic field. **d** H&E shows loss of mucosal integrity in the rectum 5.5 weeks after completing 10.9 Gy x 5 fractions QOD to a 2 cm pelvic radiation field. Scale bar = 250 μm for the low power view and 100 μm for the high-power view. **e** H&E shows severe chronic proctitis 4 months after completing 14.4 Gy x 3 fractions evenly distributed over 9 days to a 2 cm pelvic radiation field. Scale bar = 2 mm for the low power view and 300 μm for the high-power view. **f** Gross pathology of intestines 11 weeks after completing 9 Gy x 5 fractions QOD to a 2 cm pelvic radiation field. **g** H&E shows ulceration and inflammation in the cecum 11 weeks after completing 9 Gy x 5 fractions QOD to a 2 cm pelvic radiation field. Corresponds to the gross specimen shown in (**f**). Scale bar = 2 mm for the low power view and 300 μm for the high-power view. **h** Fertility of CD-1 male mice 9 months after sham treatment or 7.5 Gy x 5 fractions QOD to a 1 or 2 cm pelvic radiation field. Normal fertility is defined as ability to father a litter while housed with a female of breeding age for one month. **i** Serum testosterone from animals in (**h**). **j** Wet weight of testis and seminal vesicles from animals in panel h. P value estimate by unpaired, two-tailed t test. **k** Representative sagittal section of one seminal vesicle in CD-1 male mice 8.5 months after sham treatment or 7.5 Gy x 5 fractions QOD to a 2 cm pelvic field. H&E and blue trichrome stain showing fibrosis (black arrowhead). Note that secretions in the seminal vesicle lumen also stain blue. Scale bar = 1 mm. Inset shows higher magnification of the seminal vesicle wall. Scale bar = 100 μm. All 5-fraction regimens were delivered over a period of 9 days as shown in Table 1.

bladder or rectum. Although some prostate glands appeared smaller, there was no evidence of senescence (Supplementary Fig. 7e, f).

In some animals, the testis appeared smaller than untreated controls; however, there was no clear correlation between dose level and testis size. Germ cells are highly radiosensitive, as doses less than 1 Gy can cause prolonged azoospermia, while permanent sterility occurs in males at doses above 2 Gy[35]. The variable results we observed suggested that the testis in some mice were likely close to or within the radiation field, while others were completely shielded. Testosterone producing Leydig cells are less radiosensitive[35,36]; however, since altered testosterone levels could impact prostate cancer progression, we assessed the effects of pelvic radiation on circulating testosterone. Serum testosterone levels were not decreased in treated mice, but untreated C57BL/6 mice had very low baseline serum testosterone levels (Supplementary Fig. 7g). We therefore evaluated the effects of pelvic radiation on gonadal function in the outbred CD-1 strain, which reportedly have higher levels of serum testosterone[37]. Despite affecting fertility (Fig. 3h), we found no difference in serum testosterone or testis volume in irradiated mice relative to controls (Fig. 3i, j). Furthermore, on histologic evaluation spermatogenesis appeared to be normal (Supplementary Fig. 7h). However, the seminal vesicles, anterior prostate, and ventral prostate were on average smaller in mice treated with the 2 cm radiation field (Fig. 3j and Supplementary Fig. 7i). Histologic evaluation confirmed decreased prostate and seminal vesicle secretions and showed evidence of fibrosis (Fig. 3k and Supplementary Fig. 7j). Taken together, these data suggest that when the radiation field is targeted to the prostate, reduced fertility is not due to effects on gonadal function, but rather due to direct effects of radiation on the prostate, causing fibrosis and decreased secretory function.

**Hematologic and immunologic effects of focal pelvic radiation.** An advantage of GEMMs is that the effects of tumors and therapy on the native immune system can be studied. Having established that 7.5–9 Gy x 5 fractions delivered on an every-other-day schedule is well tolerated in mice, we next conducted a series of studies to determine the effect of SART on circulating blood cells and immune cells in spleen and lymph nodes. For all of these studies, radiation was targeted to a 2 cm circular field centered in the pelvis.

In circulation, red blood cells (RBCs) are considered to be the most radioresistant, while white blood cells (WBCs), particularly lymphocytes, are the most radiosensitive[38]. Accordingly, we found only a small decrease in RBCs after radiation that normalized 8 weeks after treatment (Fig. 4a), although the female cohort treated with 5 fractions of 9 Gy showed persistent anemia due to hematochezia. Platelet levels reached a nadir one week after radiation, and normalized by week 8 (Fig. 4a). In contrast, WBCs declined by more than 50% within 2 days of completing radiation, and the decline was primarily due to decreased lymphocytes (Fig. 4a). Neutrophil counts normalized relatively quickly, while lymphopenia persisted longer, but also resolved by week 8 (Fig. 4a).

Further characterization of circulating lymphocyte populations revealed the greatest decrease in B cells, with T cells being more radioresistant (Fig. 4b). T helper cells (Th) also appeared to be less sensitive than cytotoxic T cells (Tc), while regulatory T cells (Tregs) showed the least relative decrease relative to controls (Fig. 4b). Similar relative changes in lymphocyte subtypes occurred in spleen, albeit to a lesser extent than in blood, possibly due to the spleen being outside the radiation field (Fig. 4c, d). Axillary lymph nodes, which were also outside the radiation field, showed minimal changes in relative lymphocyte populations, while pelvic lymph nodes that were within the radiation field showed no relative differences in lymphocyte subtypes (Fig. 4c, d). It was not possible to assess absolute changes in lymphocyte populations in these organs by flow cytometry; however, lymph node weight and 2-dimensional morphometric analysis indicated that there was a trend toward smaller pelvic lymph nodes two days after radiation, the time point when maximum changes in lymphocyte populations were observed in blood (Supplementary Fig. 8a–c). At the maximum tolerated dose in females (5 fractions of 7.5 Gy) there was no measurable effect on total mass of spleen, axillary or pelvic lymph nodes (Supplementary Fig. 8a). At the maximum tolerated dose in males (5 fractions of 9 Gy) there was a small but measurable effect on total mass of spleen and axillary lymph nodes, but not pelvic lymph nodes at 2 days after completing the radiation course (Supplementary Fig. 8b).

**Efficacy of SART regimens in human-derived mouse xenografts.** Having established that pelvic SART in mice has an acceptable toxicity profile, we next evaluated efficacy in widely used xenograft models of prostate cancer. Nude mice with bilateral flank xenografts were immobilized using our custom restrainer to allow targeting of one of the flank tumors (Fig. 5a). In all cases, tumor regression was observed in the treated flank,

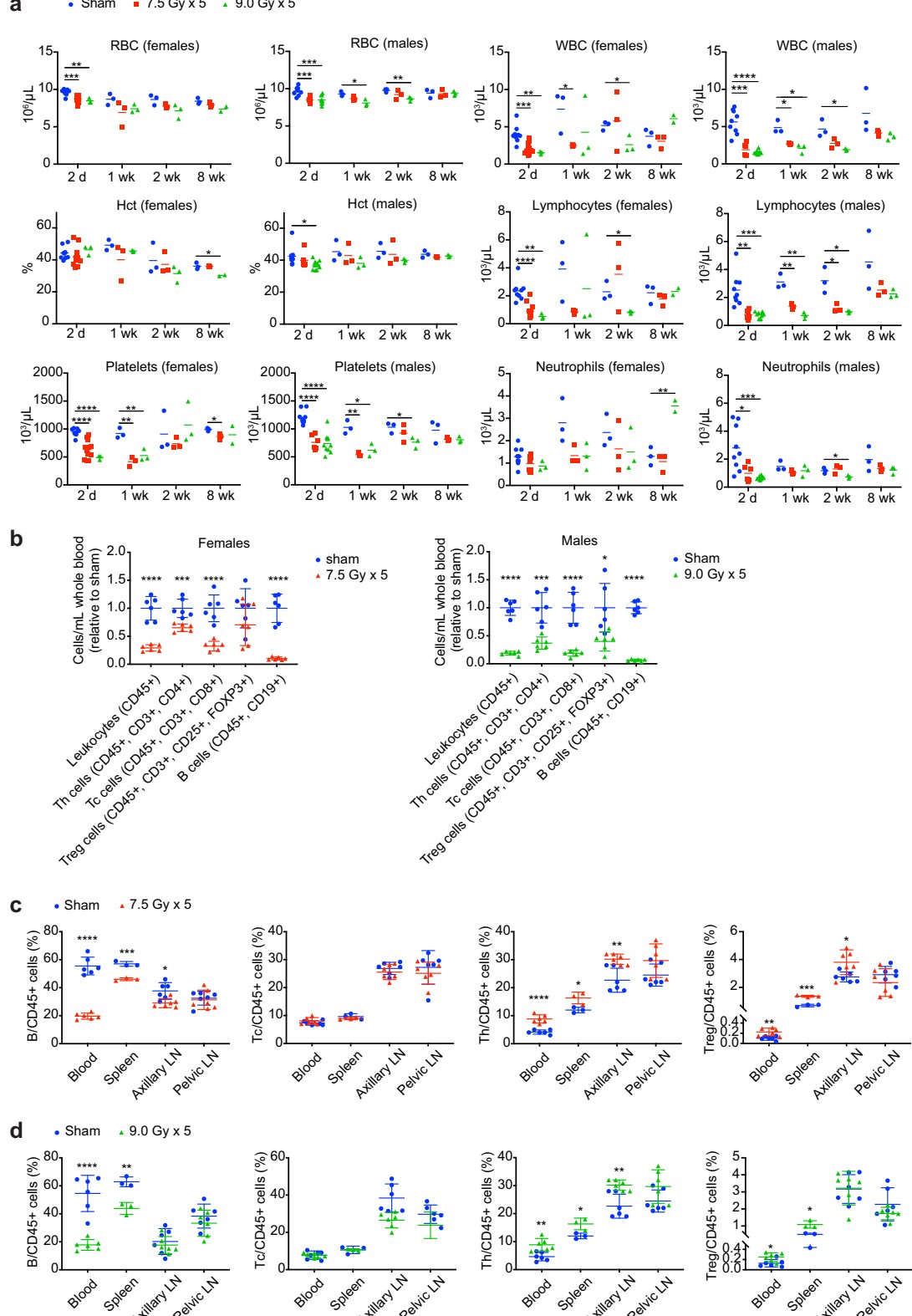

**Fig. 4 Hematologic effects of hypofractionated stereotactic ablative radiotherapy (SART) to the pelvis. a** Blood cell counts at the indicated number of days or weeks after completion of 5 fractions of radiation delivered on an every-other-day schedule or sham treatment. For females n = 3–12 for 2-day time point and n = 3 for all other time points (except n = 2 for 9 Gy x 5 group at 8-week time point due to 1 treatment-related death). For males n = 6–9 for 2-day time point and n = 3 for all other time points. Females with increased neutrophil counts at 8 weeks had chronic intestinal injury and inflammation. **b** Lymphocyte subtype counts determined by multiplexed flow cytometry in whole blood 2 days after completion of 5 fractions of radiation or sham treatment. Mean ±SD. Data are normalized to the mean of untreated controls. **c** Lymphocyte subtypes as a percent of total leukocytes 2 days after completion of radiation in females. Mean ±SD. **d** Lymphocyte subtypes as a percent of total leukocytes 2 days after completion of radiation in males. Mean ± SD. For all panels *p < 0.05, **p < 0.01, ***p < 0.001, ****p < 0.0001 by unpaired 2-tailed t test.

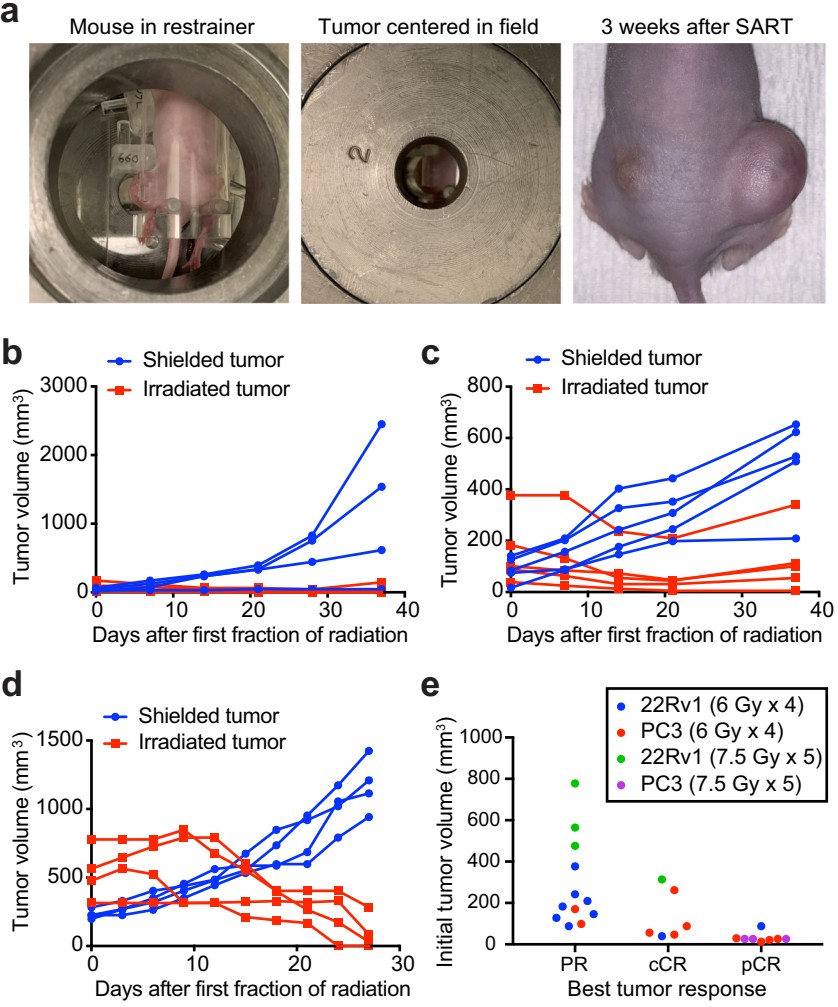

**Fig. 5 Hypofractionated stereotactic ablative radiotherapy (SART) can induce complete responses in flank xenografts. a** Nude mouse with bilateral flank 22Rv1 xenografts showing the larger flank tumor immobilized for radiation and centered in a 2 cm radiation field. Image on the right shows treatment response 3 weeks after completing 7.5 Gy x 5 fractions QOD to the left flank. **b** Growth of PC3 xenografts ($n = 4$ for shielded and $n = 5$ for irradiated) treated with 6 Gy x 4 fractions QOD to the larger flank tumor. **c** Growth of 22Rv1 xenografts ($n = 5$) treated with 6 Gy x 4 fractions QOD to the larger flank tumor. **d** Growth of 22Rv1 xenografts ($n = 4$) treated with 7.5 Gy x 5 fractions QOD to the larger flank tumor. Mean tumor volume ± SD. **e** Best tumor response by initial tumor size. PR partial response, cCR clinical complete response, pCR pathological complete response.

while the shielded tumor progressed (Fig. 5b–d). Some SART-treated tumors showed slow regrowth between 3 and 4 weeks after treatment, but most remained smaller than their pretreatment size by week 4, and some regressed completely (Fig. 5b–e). Tumors that did not regress completely still showed a reduction in proliferative index compared to untreated controls (Supplementary Fig. 9a). Eight of 14 (57%) clinical complete responders (cCR) also had a pathologic complete response (pCR), defined as no viable tumor cells present on histologic examination of the tumor implantation site (Supplementary Fig. 9b). Although PC3 xenografts were more likely than 22Rv1 xenografts to have a complete response, PC3 xenografts in our study were also smaller at time of treatment, and there was a clear trend toward better responses in smaller tumors (Fig. 5e). To determine if pathological complete responders could be considered cured, we used bioluminescence imaging (BLI) to monitor for recurrence or metastasis of luciferase-expressing PC3 xenografts after treatment with SART. BLI showed no evidence of local recurrence or metastasis 5 months after treatment, indicating that these animals indeed remained disease-free for a prolonged period after treatment (Supplementary Fig. 9c).

**Therapeutic radiation improves survival in a prostate cancer GEMM.** *PTEN* and *TP53* are commonly deleted or mutated tumor suppressors in prostate cancer, and their loss correlates with poor prognosis[39, 40]. In mice, combined deletion of *Pten* and *Trp53* in prostate epithelium is sufficient to drive tumorigenesis with short latency, resulting in rapidly progressive and lethal prostate adenocarcinoma[41,42]. Using an established GEMM (*Pten^flox/flox*; *Trp53^flox/flox*; *Pbsn-Cre* herein referred to as *Pten;Trp53^pc−/−*) we confirmed that conditional loss of Pten and p53 in mouse prostate epithelium results in locally aggressive prostate cancer that is universally fatal. Macroscopic tumors, visible by MRI, developed as early as 4 months of age (Fig. 6a) and tumors were palpable by 5 months of age. All mice died or required euthanasia due to poor body condition score between 6–8 months of age. Necropsy showed no evidence of thoracic or abdominopelvic metastasis. In some cases, pelvic lymph nodes appeared mildly enlarged, but were found to be reactive on histologic evaluation. In all cases, the cause of death could be attributed to local mass effect of the prostate tumor, resulting in genitourinary or gastrointestinal obstruction (Fig. 6b, c).

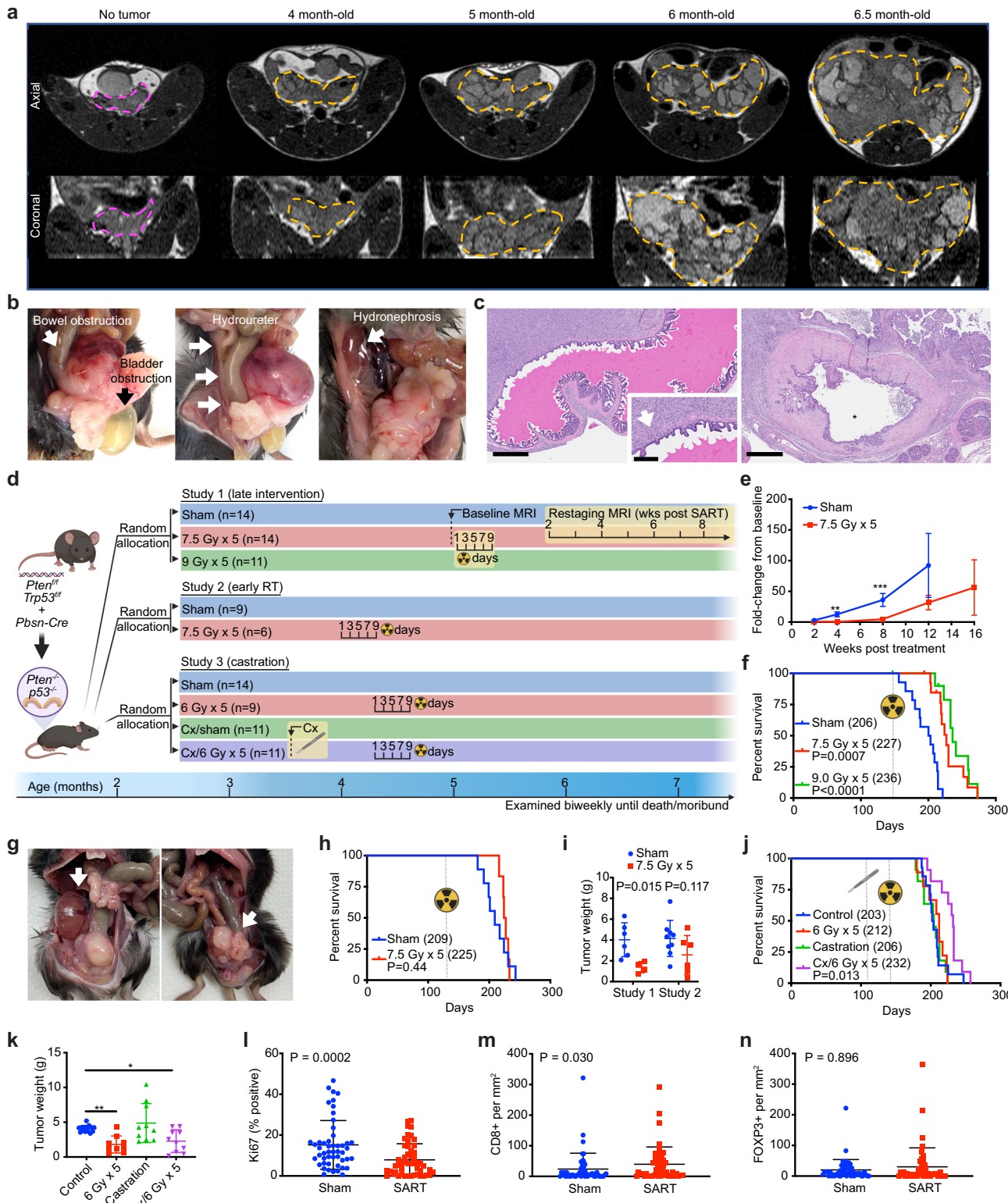

To test whether SART is effective in this prostate cancer model, *Pten;Trp53^pc−/−* male mice were randomized to 5 fractions of 0, 7.5, or 9 Gy (Fig. 6d, Study 1) at 5 months of age, a time point when tumors were clinically apparent. SART significantly delayed tumor progression assessed by serial MRI (Fig. 6e, Supplementary Fig. 10a, b), and median cancer-specific survival was extended in a dose dependent manner (Fig. 6f). Although pilot studies suggested that 5 fractions of 9 Gy was tolerable in tumor bearing mice, a few animals in the radiation

cohorts developed severe gastrointestinal toxicity requiring euthanasia (Fig. 6f). In all cases these mice had very small tumors at time of necropsy indicative of treatment effect. Necropsy of SART-treated animals that died or required euthanasia due to tumor progression showed overall smaller primary tumors; however, death was still caused by mass effect on organs adjacent to the prostate (bladder, ureter, rectum) (Fig. 6g). There was no evidence of macroscopic metastases, even in animals that survived beyond 8 months.

**Fig. 6 Stereotactic ablative radiotherapy (SART) modestly improves survival an autochthonous model of locally aggressive prostate cancer. a** Tumor progression in *Pten;Trp53$^{pc-/-}$* mice assessed by serial MRI. Representative axial view (upper panels) and coronal view (lower panels) of pelvis with prostate tumor outlined in orange. In all cases the axial and coronal slice with the largest square area of tumor is shown. A 3-month-old mouse without visible prostate tumor outlined in magenta is shown for comparison. **b** Obstruction of bladder, ureter, and rectum documented at time necropsy in untreated mice. **c** Histologic sections showing seminal vesicle invasion (arrow) and compressed bladder with small lumen (*) engulfed by tumor. Scale bar = 600 μm (main image) 200 μm (inset). **d** Graphical representation of prostate cancer GEMM and study design, including timing of interventions. Cx, castration. **e** Tumor progression assessed by MRI in study 1. n = 7 for sham cohort, n = 14 for SART cohort. Fold-change from baseline is defined as follows: (tumor volume at time of analysis—pretreatment tumor volume) divided by pretreatment tumor volume. Data is shown as mean ±SEM. See Supplementary Fig. 10a for individual tumor data points and raw tumor volume values. **$p < 0.01$, ***$p < 0.001$ by unpaired two-tailed *t* test. **f** Cancer-specific survival in study 1. Event defined as death or euthanasia due to tumor burden. Animals with cause of death other than tumor burden were censored (2 in each radiation cohort developed severe gastrointestinal toxicity requiring euthanasia). Number in parenthesis = median survival in days. *P* value estimate by log-rank test. **g** Hydronephrosis (left arrow) and rectal obstruction (right arrow) documented at time necropsy in mice treated with SART. **h** Overall survival in study 2. Number in parenthesis = median survival in days. *P* value estimate by log-rank test. **i** Tumor weight at time of necropsy in study 1 (SART at 5 months) and study 2 (SART at 4 months). Mean ±SD. *P* value estimate by unpaired two-tailed *t* test. **j** Overall survival in study 3. Number in parenthesis = median survival in days. *P* value estimate by log-rank test. **k** Tumor weight at time of necropsy in study 3. Mean ±SD. *P* value *$p < 0.05$, **$p < 0.01$ by unpaired two-tailed *t* test. **l** Percent Ki67 positive cells assessed by IHC in SART-treated tumors (9 Gy x 5) and controls from study 1 (n = 5). The time interval between treatment and tumor collection was 51–83 days. For each tumor 10 high power fields (HPF) were analyzed. Mean ± SD. *P* value estimate by Mann–Whitney test. **m** CD8+ T cells assessed by IHC in SART-treated tumors and controls as in (**l**), 10 HPF per tumor. Mean ±SD. *P* value estimate by Mann–Whitney test. **n** FOXP3+ T cells assessed by IHC in SART-treated tumors and controls as in (**l**), 10 HPF per tumor. Mean ±SD. *P* value estimate by Mann–Whitney test.

Given that tumor size at the time of radiation correlates with treatment effect (Fig. 5e) we reasoned that the marginal improvement in survival we observed was due to high tumor cell burden at time of treatment. To determine whether treating tumors at an earlier stage would improve outcomes, *Pten;Trp53$^{pc-/-}$* mice were randomized to 5 fractions of 7.5 Gy or sham irradiation (Fig. 6d, Study 2) at 4 months of age, a time point when tumors first become radiographically apparent, but are not palpable. The median tumor size by MRI at time of treatment was 52 mm$^3$ (range 25–258 mm$^3$). Surprisingly, we did not find that earlier treatment improved survival, or tumor burden, at time of death (Fig. 6h); in fact, there was a trend toward larger tumors in the early intervention cohort (Fig. 6i).

Clinical trials have demonstrated that combining radiation with androgen deprivation therapy (ADT) for prostate cancer improves outcomes over radiation alone[43,44]. To determine whether androgen deprivation improves response to RT in our model, *Pten;Trp53$^{pc-/-}$* mice were surgically castrated prior to SART (Fig. 6d, Study 3). A pilot study showed a high incidence of penile edema and prolapse in animals treated with combination therapy; therefore, castrated animals were allowed to recover for 3 weeks prior to initiating RT and the radiation dose was limited to 5 fractions of 6 Gy. Notably, at this dose, radiation alone did not improve survival despite decreasing tumor size (Fig. 6j, k). However, median survival was prolonged with combination therapy to a similar extent as high-dose SART monotherapy regimens (Fig. 6f, j). Castration alone did not improve survival or tumor burden in this model, consistent with prior reports (Fig. 6j, k)[42,45]. These results show that although *Pten;p53*-null tumors are resistant to castration, this model still recapitulates the synergy between radiation and ADT observed in human tumors.

Consistent with smaller tumors in SART-treated animals, the number of Ki67 positive cells (a marker of proliferation) was significantly lower on average in irradiated tumors (Fig. 6l). Yet, in some regions, irradiated tumors had a similar proliferation index as untreated controls, suggesting that tumor repopulation was occurring at the time tumors were collected (Supplementary Fig. 10c). We noticed that proliferation tended to be highest in areas of the tumor near normal prostate glands, and these areas also frequently showed prominent lymphocytic infiltrates (Supplementary Fig. 10d).

To further characterize tumor infiltrating lymphocytes (TILs) and understand how they are affected by RT, relevant T cell populations were examined by IHC. Although cytotoxic T cells were overall rare in tumors (in most cases <50 cells per mm$^2$), they were more abundant in regions near the tumor periphery and adjacent to normal glands (Supplementary Fig. 10e). Furthermore, levels were slightly higher on average in the core of irradiated tumors, trended higher near the tumor periphery, and were positively correlated with regions of increased tumor cell proliferation (Fig. 6m and Supplementary Fig. 10f, g). Regulatory T cells were also rare in tumors, with a few areas showing higher levels, generally where cytotoxic T cells were also more abundant (Supplementary Fig. 10h). Although regulatory T cells trended higher in irradiated tumors, unlike cytotoxic T cells, the difference was not significant (Fig. 6n). Taken together these results indicate that prostate tumors in the *Pten;Trp53$^{pc-/-}$* model are characterized by low, yet heterogeneous levels of TILs, and suggest that TIL activity may be most relevant at a time when tumor regrowth is occurring in irradiated tumors.

**Therapeutic radiation improves survival in a colorectal adenoma GEMM.** Since radiation therapy plays an important role in the management of non-metastatic rectal cancer, we also evaluated SART in a mouse adenoma model in which tumors are induced by focal deletion of *Apc* in intestinal mucosa[46]. Under endoscopic guidance, *Apc$^{f/f}$;Villin$^{CreERT2}$* mice were injected with 4-hydroxytamoxifen at a single site 2–3 cm from the anal verge to induce colorectal tumors as previously described[30]. We confirmed that tumors induced in this location could be accurately targeted with a 2 cm pelvic radiation field (Supplementary Fig. 11a, b). Although these tumors do not metastasize, they are characterized by slow, persistent growth that can ultimately lead to death due to gastrointestinal bleeding, bowel obstruction, and/or rectal prolapse. In a pilot study, a treatment regimen of two fractions of 15 Gy (BED$_3$ = 180) was effective at decreasing tumor burden, but evidence of late fibrosis was found 6 months post treatment. Thus, 5 fractions of 7.5 Gy (BED$_3$ = 130), which is well tolerated in male and female mice (Fig. 3a, Table 1), was used for treatment. After stratifying based on tumor size, mice were randomized to radiation or sham treatment and then followed for a minimum of 6 months (Fig. 7a, Study 1). While more than half of the sham-treated mice died due to bowel obstruction or required euthanasia due to severe rectal prolapse, none of the SART-treated mice developed bowel obstruction or rectal prolapse (Fig. 7b, c). At

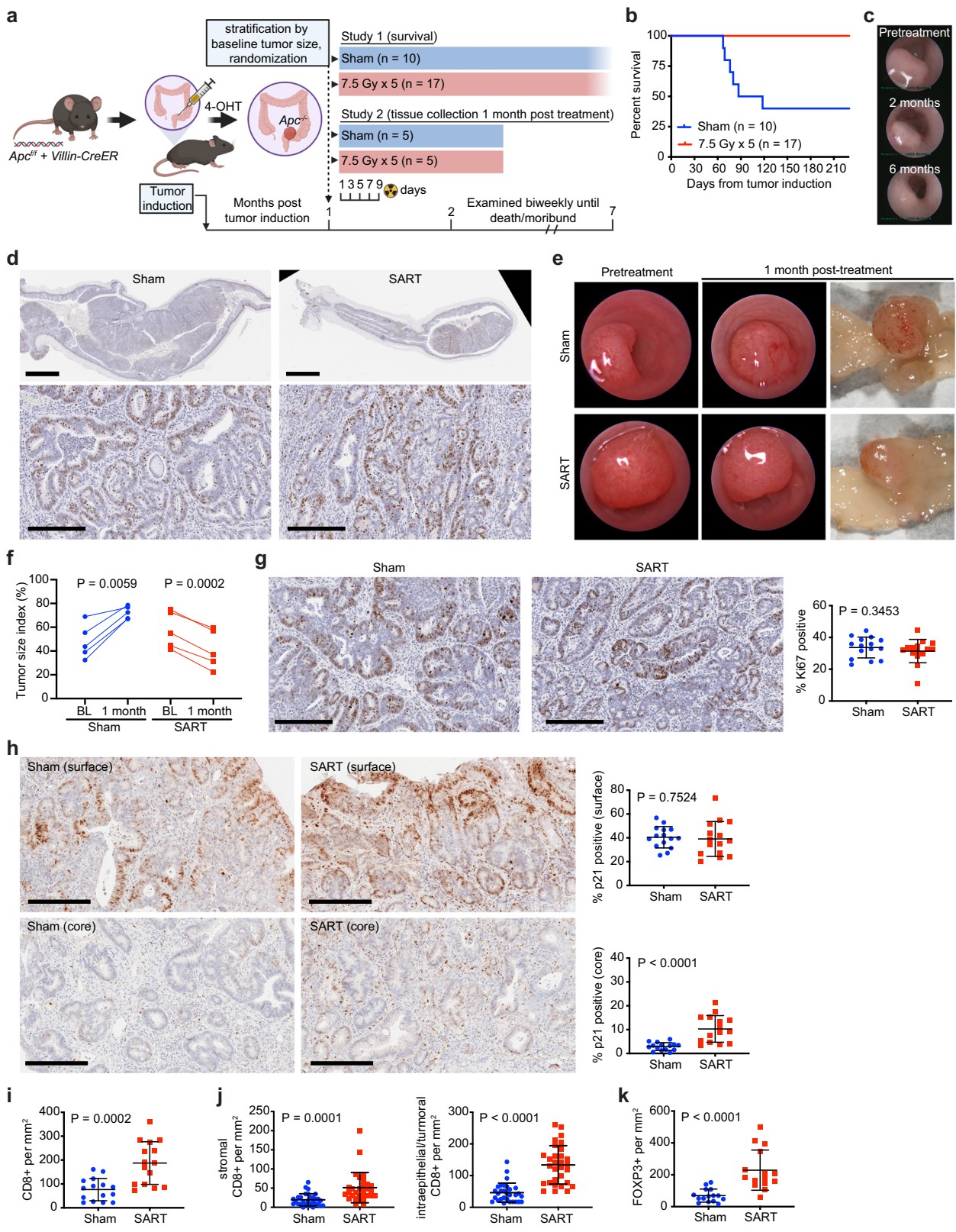

endpoint, some tumors were exophytic, while others were flatter and more infiltrative, making it difficult to compare tumor burden between individual mice; however, irradiated tumors were generally smaller and had less fluorodeoxyglucose (FDG) uptake on PET scan than controls, yet none showed complete regression, and there was no clear difference in the proliferation index (Fig. 7d and Supplementary Fig. 11c).

To determine how radiation was affecting tumors to improve survival in this model, a second study was conducted with a planned endpoint one month after treatment to allow earlier histopathologic evaluation (Fig. 7a, Study 2). Over this time interval, all irradiated tumors demonstrated regression, while all untreated tumors progressed (Fig. 7e, f). Interestingly, no difference in the proliferation index, as assessed by Ki67

**Fig. 7 Stereotactic ablative radiotherapy (SART) improves survival in an autochthonous model of colorectal adenoma. a** Graphical representation of colorectal cancer GEMM and study design, including timing of interventions. Study 1 was conducted with approximately equal numbers of male and female mice that were 2–3 months of age at time of tumor induction. Study 2 involved only male mice. 4-OHT, 4-hydroxytamoxifen. **b** Overall survival in study 1. **c** Representative endoscopic images of rectal adenoma prior to treatment and at the indicated time points after SART. **d** Representative Ki67 IHC of a rectal adenoma 7 months after radiation (8 months after tumor induction). A tumor from an untreated animal, two and a half months after tumor induction, is shown for comparison. Scale bar = 2 mm (low power view), 0.2 mm (high power view). **e** Representative endoscopic and gross images of a SART-treated tumor and untreated control from study 2. **f** Size of individual tumors from study 2 as evaluated by endoscopy at baseline (BL) and 1-month post SART. $n = 5$ for both treatment groups. $P$ value estimate by paired, two-tailed $t$ test. **g** Representative Ki67 IHC images and proliferative index of tumors from study 2. Three high power fields per tumor were assessed. Scale bar = 0.2 mm. $P$ value estimate by Mann–Whitney test. **h** Representative p21 IHC images and quantitation of p21 positive cells at the surface and core region of tumors from study 2. Three high power fields per region/tumor were assessed. Scale bar = 0.2 mm. $P$ value estimate by unpaired, two-tailed $t$ test. **i** Quantitation of cytotoxic T cells assessed by IHC in tumors from Study 2. Three high power fields per tumor were assessed. $P$ value estimate by unpaired, two-tailed $t$ test. **j** Quantitation of cytotoxic T cells assessed by IHC in stromal vs intraepithelial/tumor compartments of tumors from Study 2. Six high power fields per tumor were assessed. $P$ value estimate by unpaired, two-tailed $t$ test. **k** Quantitation of regulatory T cells assessed by IHC in tumors from Study 2. Three high power fields per tumor were assessed. $P$ value estimate by unpaired, two-tailed $t$ test.

immunostaining, was found in irradiated tumors (Fig. 7g). We also found no evidence of apoptosis (Supplementary Fig. 11d). Based on these findings, we hypothesized that SART-treated tumors were undergoing proliferation arrest. Since, *Apc*-null adenomas have intact p53 signaling, and p21 (CDKN1A) is key mediator of p53-mediated growth arrest in response to DNA damage in colorectal cancer cells[47,48], we investigated p21 expression by IHC in adenomas 1-month post treatment. Interestingly, we found substantial spatial heterogeneity in p21 expression in the adenomas. Tumor cells near the luminal surface showed high levels of p21 expression, and there was no difference between SART-treated tumors and controls (Fig. 7h). In contrast, adenoma cells deeper in the tumor had low p21 expression; however, the core of irradiated tumors had a significantly higher fraction of p21 positive cells, the majority of which were in the stroma (Fig. 7h).

Immune cells can also exhibit anti-tumor effects independent of apoptosis[49]. We therefore quantified TILs in adenomas 1 month after treatment, a time point at which treated tumors showed regression. IHC analysis showed that cytotoxic T cells were present in intraepithelial and stromal compartments, while regulatory T cells were restricted to the stromal compartment, and both were increased in irradiated tumors (Fig. 7i, j and Supplementary Fig. 11e, f). These data suggest that SART affects tumor stroma and immune cells in a way that could contribute to therapeutic effects of radiation in this autochthonous colorectal cancer model.

## Discussion

Disease recurrence after curative therapy remains a major challenge in the clinic, and both treatment-resistant persister cells and an immunosuppressive tumor microenvironment have been implicated as contributors to this problem. GEMMs provide an opportunity to examine intrinsic determinants of tumor response and evaluate the contribution of extrinsic factors[3–5]. Here we show that potentially curative RT regimens can be employed to treat GEMMs of prostate and colorectal cancer, resulting in delayed tumor progression and changes to the tumor microenvironment. Importantly, we show that survival is improved in both models and that treatment-related toxicity does not preclude assessment of tumors with slow growth kinetics. Furthermore, we observed synergy between radiation and androgen deprivation as has been demonstrated in human prostate cancer[43,44].

While SART achieved pathologic complete responses in xenograft models, none of the GEMM tumors regressed completely, and in the case of the locally aggressive *Pten;Trp53*$^{pc-/-}$ prostate cancer model, survival was limited due to local progression. A recent meta-analysis of 18 randomized trials showed

that local failures after prostate RT occur in at least 20% of patients with high-risk disease, and 7.8% of patients with intermediate-risk disease[50]. Although these patients all received conventionally fractionated RT, clinical outcomes, including local failure rates, are similar for hypofractionated and conventionally fractionated RT[51]. Collectively these data argue that tumor biology plays an important role in defining the RT response in prostate cancer. Although our study was not designed to examine mechanisms of radiation resistance, the finding that treating autochthonous tumors at an earlier stage and smaller size did not improve outcomes is notable. In particular, the observation that there was a trend toward larger tumors in *Pten;Trp53*$^{pc-/-}$ mice treated at an earlier stage was unexpected. This contrasts with xenograft models where tumor control probability was associated with initial tumor size. Furthermore, upon prostate tumor regrowth there was heterogeneity in proliferative index, with some areas showing similar proliferation as untreated tumors. One potential explanation is that these tumors harbor a constant number, rather than a fixed percentage, of radioresistant clones, such that treatment-resistant cells in animals treated at an earlier stage have more time to grow and result in larger tumors. Another possibility is that smaller tumors have underlying biological differences, such as slower proliferation rate, making them less likely to respond to radiation. With regard to molecular mechanisms of radioresistance, although it has been shown that some GEMMs do not acquire additional mutations other than the engineered driver mutation(s)[52], other GEMMs are characterized by many non-synonymous mutations[53], raising the possibility that genomic alterations could render some clones more radioresistant. However, epigenetic alterations, transcriptional programs, and extrinsic factors can also define a radioresistant cell state[54–56]. Additional work is needed to determine whether genomic alterations or transcriptionally-defined cancer cell states underlie treatment resistance in this model.

In the rectal adenoma model, tumor associated stroma was markedly altered by SART treatment, raising the question of whether radiation effects on stroma contribute to therapeutic efficacy. Tumor stroma consists of multiple cell types, including cancer associated fibroblasts, immune cells, and vascular cells, all of which can be affected by radiation and also interact with each other in complex ways[57–60]. Future studies are needed to better define how radiation alters tumor stroma to shape response to therapy.

To conduct these studies, we developed a stereotactic RT platform designed to immobilize and treat extracranial tumors in mice. Apart from cost and ease of use, an advantage of the platform described here is that animals are conscious during treatment, thus avoiding repeated exposure to anesthesia that could affect animal health and alter the efficacy of radiation

therapy[61,62]. Moreover, we show that due to the predictable anatomy of laboratory mice, pretreatment imaging is not required to target autochthonous tumors with focal RT provided that appropriate immobilization and stereotactic technique are used. For practical reasons our study employed a parallel opposed beam arrangement; however, repositioning the restrainer in a fixed source irradiator, or use of a gantry-based irradiator, would allow more conformal treatment of deep tumors and better mimic human SABR dosimetry, albeit with longer overall treatment time. In clinical practice, image guidance has allowed for more conformal treatment of tumors, and while image guidance could in theory allow for more precise targeting of RT in mice, a major challenge for targeting autochthonous tumors with image-guided RT is that they are frequently infiltrative and margins cannot be accurately defined even with high resolution CT or MRI. Thus, approaches to better determine the extent and spatial distribution of microscopic disease in autochthonous tumors is needed in order to use image guidance for more conformal treatment of these tumors with curative intent. Another challenge is that available image-guided irradiators for small animals use kilovoltage x-rays of relatively low effective photon energy, which are less skin sparing, may have less repairable DNA damage from the higher linear energy (LET) component, and have a substantial photoelectric effect with implications for higher bone marrow dosing and enhanced damage to the hematopoietic system[63,64]. Therefore, there are also some dosimetric advantages to Cesium-137 gamma irradiators, such as the one employed in this study, which have a higher effective photon energy that is closer to photon energies used in the clinic.

The acute and chronic effects of whole body radiation in mice is well documented (summarized in[65]), yet the tolerance of individual organ systems to clinically relevant SABR regimens has not been established. We found that mice tolerate hypo-fractionated (5-fraction) pelvic RT regimens using similar doses employed in the clinic for definitive treatment of prostate cancer. In male mice, no gastrointestinal toxicity was observed at doses up to 7.5 Gy per fraction delivered every other day and targeted to a 2 cm pelvic radiation field; however, when the tolerance dose was exceeded, acute rectal toxicity (ulceration, colitis) resulted in mortality around 5 weeks after treatment. In contrast female mice developed mild-moderate GI toxicity, primarily bowel enlargement and edema involving the ilium, at a dose of 7.5 Gy per fraction. One reason for the sex differences may be the larger volume of small intestine within the radiation field of female mice, as females tolerated 5 fractions of 9 Gy when the field size was reduced. In summary, we find that SABR regimens used to ablate tumors in the clinic are well tolerated when targeted to the pelvis in mice. When the maximum tolerated dose is exceeded, acute lethal injury is due to colorectal toxicity. For a 5-fraction regimen, doses in the 7.5–9 Gy per fraction range can result in subacute to chronic intestinal toxicity. In contrast, distal colon, rectum, bladder, prostate, and seminal vesicles are less affected at these doses, yet in some cases develop fibrosis as a late effect of radiation.

Although the toxicity studies primarily involved C57BL/6 J mice, which are of intermediate sensitivity with regard to whole body radiation effects[66], a subset of studies were also conducted in outbred CD-1 mice with similar results. However, caution is advised in extrapolating these results to inbred strains that are more sensitive to ionizing radiation[67], or animals with underlying comorbidities, genetic alterations, or are undergoing concurrent therapy, where toxicity may be increased. In these situations, it is advisable to begin with regimens in the $BED_3 = 90$–130 range with a maximum field size of 2 cm and minimum body weight of 20 g, and the field size should be decreased for smaller mice.

Regarding acute systemic hematologic effects, our findings recapitulate some findings involving single-fraction whole body radiation and show that similar changes occur with clinically relevant SABR regimens[38,68,69]. Consistent with reported differences in radiosensitivity among blood cell lineages[70–73], the highest relative decrease and most prolonged effect was observed for lymphocytes. B cells were the most sensitive lymphocyte population, while the relative radiosensitivity amongst T cells was Tc > Th > Treg. An open question is whether the effects we observed on peripheral blood cells are caused by direct effects of radiation on circulating cells or are due to depletion of hematopoietic progenitors in the radiation exposed bone marrow. Two lines of evidence suggest the former. First, the timing of the observed changes relative to the half-life of circulating blood cells is inconsistent with marrow suppression; second, the fraction of functional bone marrow within a 2 cm pelvic radiation is relatively small and should be compensated for by bone marrow outside the radiation field[74]. While it is possible that background radiation (estimated to be 5-8% of the prescription dose) could affect lymphocyte populations outside the radiation field, such doses are unlikely to affect erythrocyte, granulocyte, and megakaryocyte precursors in the bone marrow[73]. Interestingly, similar effects on peripheral blood cells have been observed in humans receiving pelvic RT, where again the timing of changes are more suggestive of effects on peripheral circulating blood cells rather than bone marrow suppression[75,76].

Lymphocyte subsets in the spleen and lymph nodes within and outside the radiation field were also evaluated. Although outside the radiation field, the spleen showed similar, albeit less pronounced, changes in lymphocyte populations compared to whole blood, namely a relative decline in B cells and increase in Th and Treg cells. However, no consistent shift in lymphocyte populations was observed in axillary or pelvic lymph nodes. Taken together these data suggest that lymphocytes in lymph nodes are less radiosensitive than circulating lymphocytes, and that some lymphatic tissues outside the radiation field may undergo transient lymphodepletion. Although morphometric analysis suggested that radiation affected in-field lymph nodes, we could not demonstrate this based on lymph node tissue weights, possibly due to the small size of pelvic lymph nodes and considerable variation in lymph node weight between animals at baseline. Interestingly, a small but statistically significant decrease in weight of spleen and lymph nodes outside the radiation field occurred in mice treated with 5 fractions of 9 Gy but not 7.5 Gy. While these changes may have been caused by low dose background radiation, an alternate possibility is that lymphocyte populations outside the radiation field were mobilized in response to peripheral lymphocyte depletion.

In the era of immunotherapy, how radiation and other cytotoxic therapies interact with the immune system is an area of increasing interest[77,78]. In preclinical models, radiation has been shown to have both immunostimulatory and immunosuppressive effects[60,79,80]. The GEMMs of prostate and rectal cancer used in this study have variable amounts of TILs with notable spatial heterogeneity. Interestingly, TILs were increased after SART in both models, implying that the immune system may help define how autochthonous tumors respond to RT. This increase was observed even at time points with transient reduction in circulating lymphocytes. In the rectal model both Tc and Treg cells were increased, while in prostate only Tc cells showed a statistically significant increase. In the prostate model Tc cells were lower in the tumor core than at the periphery, but were specifically increased in the core following SART, suggesting that radiation could induce deeper penetrance of Tc cells into the tumor core, a possibility that deserves further investigation. Overall, these findings are in line with prior reports of increased

intratumoral Tc and Treg cells after radiation in syngeneic mouse xenograft models, and increased survival of TILs compared to lymphocytes in circulation and lymphoid tissues[81–83]. Importantly, our data show that this phenomenon also occurs in autochthonous tumors, which have an intact tumor stroma that can impact TIL recruitment and may more closely resemble the immune cell microenvironment found in human tumors[84] In fact, the response to radiation in combination with immune checkpoint inhibitors can vary dramatically between transplanted and autochthonous tumor models, arguing that tumor environment is an important factor in determining response to radiation and immunotherapy[53].

In summary, we show that clinically relevant SABR regimens can be delivered safely to mouse cancer models, but in some cases are unable to cure autochthonous tumors, mimicking the behavior of some tumors in the clinic. Thus, these models can be used to examine how cancer evades radiation treatment and develop approaches to improve therapeutic efficacy.

## Data availability

All data supporting the findings of this study are available within the manuscript and supplementary files. Source data for Figs. 1–7 and Supplementary Figures 1–11 can be found in the separate source data file provided with this manuscript (Supplementary Data 1).

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

## Acknowledgements

We thank Gerry Rondeau for constructing the lead shields, collimators, and restrainers; Xing-Qi Lu for assisting with dosimetry; Michael Brown and Alicia Caron for assisting with histology; Ellen Buckley for assisting with hematology; Dr. Pier Paolo Pandolfi for providing the Pten;Trp53pc-/- model and Dr. Sean Clohessy for advice on husbandry for this model. Graphical representations of the GEMM models and timelines were created with Biorender. DRS acknowledges funding from DF/HCC SPORE in Prostate Cancer Training Award P50 CA090381-15, the Harvard University KL2/Catalyst Medical Research Investigator Training award TR002542, and the Joint Center for Radiation Therapy Foundation. MVH acknowledges funding from Ludwig Center at MIT, MIT Center for Precision Cancer Medicine, Stand up to Cancer, Emerald Foundation, Howard Hughes Medical Institute faculty scholar award, and NIH grants P30CA1405141, R35CA242379, P50CA090381. JR acknowledges funding from DOD grant W81XWH-20-1-0203, Duke-NC State Translational Research Grant, and NIH grants R37CA259363, R21CA256414, R21DK125911, R41EB032693, R01CA254108, R01CA256530, and R01CA244359 (JR). DRS and LED were supported by NCATS/NIH Award UL1 TR002541 to Harvard Catalyst and The Harvard Clinical and Translational Science Center and financial contributions from Harvard University and its affiliated academic health care centers.

## Author contributions

Designing research studies: D.R.S., J.R., J.D.D., S.R.F., M.V.H. Conducting experiments: D.R.S., J.R., I.M.T.G., A.S., C.L.W. Method Development: D.R.S., J.D.D., J.R., A.S., C.L.W., M.R.C., K.C., C.C., E.H. Acquiring Data: D.R.S., J.R., I.M.T.G., A.S., C.L.W., W.H., H.H.M., Analyzing Data: D.R.S., I.M.T.G., A.S., C.L.W., M.H., W.S.K., M.R.C. Statistics review: L.E.D. Supervision: M.V.H., M.A.S., O.H.Y., K.D.W. Manuscript writing: D.R.S., M.V.H. Manuscript review & editing: J.R,. S.R.F., J.D.D., I.G., A.S.

## Competing interests

M.G.V.H. discloses that he is a scientific advisor for Agios Pharmaceuticals, iTeos Theapeutics, Sage Therapeutics, Droia Ventures, and Auron Therapeutics. These competing interests do not affect the current study. The other authors declare that they have no competing interests.
