## [Peer Review File · Communications Medicine]

Reviewers' comments:

Reviewer #1 (Remarks to the Author):

The authors established the methods for delivering radiotherapy to genetically engineered mouse models (GEMMs) of primary prostate and colorectal cancer and evaluated the tumor response to SABR. They reported that SABR can be delivered in this setting and the changes in the tumor stroma and immune environment that led to tumor control.

Immune responses to radiotherapy are an area of great interest nowadays, and GEMMs are rarely used to model this. Therefore, from this perspective, I believe that the results of the present study have the potential to impact this scientific field.

However, this manuscript has a considerable number of concerns as listed below.

Therefore, the reviewer thinks that the manuscript is not acceptable for publication in Communications Medicine in its current form.

Major concerns:

The Results and Discussion section is redundant and somewhat confusing. There are a variety of dose/fraction regimens that were used, so please think about how to present them in a way that is easy to understand. For example, responses and side effects should be listed in a table for each regimen. It would be preferable to update the entire manuscript, including this example.

Throughout the text, the irradiation dose/fractionation method and the time period between irradiation and analysis should be clarified. Explanations of this important radiobiological information are missing from several figures.

In some regimens in this study, the irradiation period appears to be set as every other day instead of every day. What is the rationale and purpose for this duration of irradiation? Is the main reason to avoid side effects? In this respect, although the irradiation dose reproduces clinical SABR, it does not strictly reproduce a clinical model that basically irradiates patients every day, and therefore the expression "SABR regimens used to ablate tumors in the clinic" in line 245 sounds like an overstatement.

Fig 6E, S6A: The y-axis in Fig 6E should be explained in the text or legends. Also, in these figures, treatment should have started at 5 months of age, i.e., when mice have clinically apparent tumors. Is it correct that the tumor volume is 0 cm³ in 'week 0' of the weeks after treatment? Or are the tumors too small at week 0 and just look like 0 cm³ on the figure?

Fig. 6I: Since tumor volume at the beginning of RT also affects the response rate even in clinical settings, and early intervention is generally beneficial, it is interesting to discuss why there was a trend toward larger tumor volume in study 2 compared to study 1. Since there is no difference in volume in the sham group, this result seems to imply that "early initiation of RT promotes tumor growth". Is this a phenomenon specific to GEMMs in this study? The reasons for this result should be added in the Discussion section.

Fig. 6L: Since the time from RT is thought to affect Ki67, information on the dose and timing of specimen collection should also be helpful for readers.

Fig. 6M: The authors stated that CD8+ TILs were overall rare and their distribution varied by region. Therefore, the regions evaluated in this figure should be clarified. Specifically, it would be informative to divide the regions into two groups: tumor periphery and adjacent to normal glands, where CD8+ TILs are more abundant, and other regions.

Line 378: Since relatively few Ki67-positive cells were observed in the SABR group (Fig. 6L), while CD8+ TILs were significantly higher in the SABR group (Fig. 6M), does this contradict the statement "TIL activity may be most relevant at a time when tumor regrowth is occurring in irradiated tumors"? Alternatively, could authors consider showing co-staining of Ki67 and CD8 TILs that would support this statement?

Fig. 7I, J: If the distribution of CTLs and Tregs is different, it would be interesting to see the differences in intraepithelial and stromal distribution, respectively. It is important to discuss the effects of RT on the immune environment, which is currently the subject of many studies, in GEMMs.

Line 437: As shown in Figure 6H, since there was no survival benefit in the early treatment model (study 2), this part seems to be somewhat overstated.

Line 457: Even if small autochthonous tumors have radioresistant persister cells, it wouldn't be able to explain the reason why the effect of radiotherapy seen in larger tumors is diminished in small tumors. Instead, is there a biological basis for the radioresistance particular to small tumors? Or perhaps it only reflects a rise in deaths brought on by side effects? Please discuss these issues.

Minor:

Please unify the notation of Ki67 and Ki-67 in entire the manuscript.

Line 421: Is it "one" instead of "on"?

Line 532: The axillary LNs seem to show a significant increase in Th in both males and females in the irradiated group. This may not be consistent with the description in the Discussion.

Fig. S3D: The explanation seems inadequate. The organs and doses for each image could be clearly indicated in the figure or legends.

Fig S3H: Does red dots show 7.5 x 5? Furthermore, the entire manuscript should clarify which RT schedule is for 7.5 Gy x 5 fr.

Fig. S3J: It would be better to show images of the pelvis and lower abdomen separately, as they appear to have different irradiated fields.

Line 192: The expressions of "DL4" in the manuscript are confusing. Please describe the dose and the fractions.

Fig S4C: Next to images, it is easier to understand if it is also represented graphically.

Fig 5A: It would be helpful to be described the irradiation dose.

Fig 6: The order is back and forth in the manuscript, so correct it.

Reviewer #2 (Remarks to the Author):

This is an interesting large scale study on the use of hypofractionation on genetically modified mice. I find the results and conclusions very interesting and useful to the field. I also think it was an excellent choice to deliver these doses without anesthesia, as using anesthesia can influence the radiation response. The dosimetry has also been well worked out. I was hoping for a couple of points of clarification regarding the scope of the study. In addition, I have some minor suggestions for improvement.

The largest criticism of this paper that I have is the premise that this is SABR. In my mind, SABR is not only dose escalation, but also high conformity. All treatments described in this study are done with a Cs-137 source in a simple AP-PA beam arrangement. In my mind, this is a hypofractionation study, not SABR. That being said there are still a great deal of important findings in this paper. Understanding both the treatment response to the target area and the dose tolerances to OARs is vital information for using this technique on genetically modified mice. I think that reframing this as a study on hypofractionation would be more representative of the study described here. It should also be noted here that Cs-137 sources are becoming less popular in the field, as small animal irradiators based on x-ray tubes are being phased in. Expanding this study using a small animal irradiator with onboard imaging would greatly improve the calculated dose statistics that are used in this study.

Minor Points:

- 1) In terms of the dose calculation presented in this paper, how was the CT density table established?
- 2) This manuscript would greatly benefit from a table summarizing the dose regimens tested, and the associated tumor control / normal tissue toxicity. Please add this to the main manuscript.
- 3) Ki67 expression is featured in this paper but not explained. Can you please provide context for why this is important?
- 4) Please provide more details on how the film dosimetry was conducted. Absolute film dosimetry can be tricky if not done very carefully. Typically new calibration curves are recommended with each new batch of film due to inter-batch variability.

RESPONSE TO REVIEWER COMMENTS (Reviewer comments in blue)

Response to Reviewer #1

The authors established the methods for delivering radiotherapy to genetically engineered mouse models (GEMMs) of primary prostate and colorectal cancer and evaluated the tumor response to SABR. They reported that SABR can be delivered in this setting and the changes in the tumor stroma and immune environment that led to tumor control.

Immune responses to radiotherapy are an area of great interest nowadays, and GEMMs are rarely used to model this. Therefore, from this perspective, I believe that the results of the present study have the potential to impact this scientific field.

However, this manuscript has a considerable number of concerns as listed below. Therefore, the reviewer thinks that the manuscript is not acceptable for publication in Communications Medicine in its current form.

We thank the Reviewer for their thoughtful critique of the manuscript and for their insightful comments. We think that addressing the points raised has improved the manuscript and hope that these revisions have addressed the Reviewers concerns.

Major concerns:

The Results and Discussion section is redundant and somewhat confusing. There are a variety of dose/fraction regimens that were used, so please think about how to present them in a way that is easy to understand. For example, responses and side effects should be listed in a table for each regimen. It would be preferable to update the entire manuscript, including this example.

We thank the Reviewer for pointing this out, and apologize for any confusion as we strive to present our findings in a way that is as accessible as possible. When preparing the manuscript, we had considered how best to present the dose/fractionation regimens. We wanted to compare/contrast courses with similar effective dose yet different number of fractions, and present this complex information in a way that is clear for a general audience that includes non-radiobiologists. We therefore settled on referring to the regimens by dose level, but recognize that having this information in a Supplementary Table might be missed by readers. Thus, in the revised manuscript we have moved this information to a Table in the main manuscript (Table 1 in the revised manuscript). Where possible, we have also updated the text to refer to regimens by fractional dose as well as the number of fractions and schedule. We hope the Reviewer agree that this improves the data presentation in the revised manuscript.

Throughout the text, the irradiation dose/fractionation method and the time period between irradiation and analysis should be clarified. Explanations of this important radiobiological information are missing from several figures.

The Reviewer raises a good point, and we agree that it is critical to clearly present specifics about dose/fractionation as well as timepoints of analysis relative to treatment. In the revised manuscript this information has been clarified throughout the text and figure legends. We also moved Supplemental Table 1 to the main manuscript (now Table 1 as discussed above), which we think will help avoid confusion about the dose, fractionation, and schedule used in each experiment.

In some regimens in this study, the irradiation period appears to be set as every other day instead of every day. What is the rationale and purpose for this duration of irradiation? Is the main reason to avoid side effects? In this respect, although the irradiation dose reproduces clinical SABR, it does not strictly reproduce a clinical model that basically irradiates patients every day, and therefore the expression "SABR regimens used to ablate tumors in the clinic" in line 245 sounds like an overstatement.

The Reviewer makes an excellent point as time between fractions is an important variable that is frequently modulated in clinical practice. In pilot experiments we found that an "every-other-day" regimen was better tolerated than a daily regimen with regard to acute gastrointestinal toxicity, and this led us to use the "every-other-day" regimen for all tumor studies. Included in the manuscript is a formal comparison of daily vs every-other-day schedules at a dose of 7.5 Gy x 5 fractions (see Table 1 in the revised manuscript). While daily SABR treatment is used for many treatment sites in the clinic, treating every other day, or otherwise extending the time period between first and last fraction is also common in clinical practice. For example, in our clinical practice, we deliver 5-fraction prostate SABR over a period of 7 days, which we find reduces incidence of acute toxicity. In fact, the ASTRO/ASCO/AUA guideline recommends delivering prostate SBRT every other day rather than daily (PMID 30316897), a recommendation that is supported by phase II data (PMID 18755555). In our department, lung SABR is also generally delivered every other day, a schedule that is favored 3:1 over daily treatment in the United States (see ASTRO 2021 abstract 2918, IJROBP Volume 111, Issue 3, Supplement, 2021, Page e444,). Daily lung SABR treatment has been associated with higher rates of acute toxicity than non-daily treatment regimens (e.g. PMID 23993401). Thus, non-daily SABR regimens are common in clinical practice, have been shown to reduce the risk of acute toxicity in the clinic, and were also associated with decreased acute toxicity in the mouse studies presented here. It is likely that other schedules that treat daily or every other day during the week, but with weekend breaks, may also be feasible and would be more practical for some researchers, and is something we will test in the future. Of note, in response to a suggestions from Reviewer #2, in the revised manuscript we now refer to our treatment as stereotactic ablative radiotherapy (SART), which we hope will avoid any misconceptions with how the dose distribution and conformality of our preclinical radiotherapy platform compares to clinical delivery of SABR.

Fig 6E, S6A: The y-axis in Fig 6E should be explained in the text or legends. Also, in these figures, treatment should have started at 5 months of age, i.e., when mice have clinically apparent tumors. Is it correct that the tumor volume is 0 cm³ in 'week 0' of the

weeks after treatment? Or are the tumors too small at week 0 and just look like 0 cm³ on the figure?

We apologize that the axis for this figure was confusing. We graphed the data in Fig 6E as fold-change from baseline” where fold-change is defined as

$$\frac{\text{final volume} - \text{initial volume}}{\text{initial volume}}$$

This definition has been added to the figure legend. Of note, Fig 6E and S6A are the same data. The main Figure 6E was chosen for clarity of presentation. The Supplementary Figure S6A was included for transparency to show unnormalized individual data points. We now reference the Supplementary Figure in the main figure legend for readers to easily find the raw data. Tumor volume was defined by contouring the GTV on axial MRI slices (slice thickness = 0.5 mm). The average tumor volume at “week 0” was 64 mm³ = 0.06 cm³. This is approximately equivalent to a sphere with diameter of 5 mm. For a representative tumor with a beginning volume of 0.06 cm³ and end volume of 3 cm³, the fold-change is 49. A magnified view of the y axis for the graph in S6A is shown below. We think this graph does not add to interpretation of the data presented; however, it could be added to the Supplementary Figure if the Reviewer or Editor thinks it would improve the manuscript. Additionally, we are happy to present the data in the main figure differently if the Reviewer or Editor has another preference.

Fig. 6I: Since tumor volume at the beginning of RT also affects the response rate even in clinical settings, and early intervention is generally beneficial, it is interesting to discuss why there was a trend toward larger tumor volume in study 2 compared to study 1. Since there is no difference in volume in the sham group, this result seems to imply that "early initiation of RT promotes tumor growth". Is this a phenomenon specific to GEMMs in this study? The reasons for this result should be added in the Discussion section.

Indeed, this is interesting and was not what we expected to find. However, we do not

think that early initiation of RT promotes tumor growth because the tumors in study 2, while trending larger than treated tumors in study 1, are still smaller than untreated controls. In considering potential explanations of the data, we think that it is possible that a trend toward larger tumors in study 2 could reflect the longer interval between treatment and death. For example, if at the time of treatment there were a similar number of treatment-resistant cells present in tumors from study 1 and study 2, tumors in study 2 could be larger because the treatment-resistant cells had a longer time to grow. This would imply that untreated *Pten;Trp53^{pc/-}* tumors harbor a constant number of treatment-resistant cells over time rather than having a fixed percentage of treatment-resistant clones. This hypothesis deserves further investigation in future studies. We have added a discussion of this to the Discussion section of the revised manuscript (page 21).

Fig. 6L: Since the time from RT is thought to affect Ki67, information on the dose and timing of specimen collection should also be helpful for readers.

This information was added to the relevant figure legend in the revised manuscript.

Fig. 6M: The authors stated that CD8+ TILs were overall rare and their distribution varied by region. Therefore, the regions evaluated in this figure should be clarified. Specifically, it would be informative to divide the regions into two groups: tumor periphery and adjacent to normal glands, where CD8+ TILs are more abundant, and other regions.

For the analysis shown in Fig 6M, 10 HPFs were selected at random throughout the tumor, avoiding areas of necrosis and areas containing normal glands (see examples below). This has also been clarified in the methods section of the revised manuscript (page 41). In addition, we performed the additional analysis requested, wherein we separately quantify CD8+ TILs in tumor regions adjacent to glands at the periphery of the tumor. In these regions (which make up a small percentage of the total tumor volume) we find overall higher number of CD8+ TILs in both treated tumors and controls, and a trend toward more TILs in treated tumors relative to controls (see Supplemental Figure 6F in the revised manuscript).

HPFs in representative sham tumor for analysis shown in Fig 6M

HPFs in representative RT-treated tumor for analysis shown in Fig 6M

Line 378: Since relatively few Ki67-positive cells were observed in the SABR group (Fig. 6L), while CD8+ TILs were significantly higher in the SABR group (Fig. 6M), does this contradict the statement " TIL activity may be most relevant at a time when tumor regrowth is occurring in irradiated tumors"? Alternatively, could authors consider showing co-staining of Ki67 and CD8 TILs that would support this statement?

We observed that proliferation in treated tumors tended to be highest in areas of the tumor near normal prostate glands (Supplemental Figure 6D) and these areas also frequently showed more prominent lymphocytic infiltrates (Supplemental Figure 6E). Unfortunately, co-staining is not possible because both primary antibodies are rabbit monoclonals; however, we have now quantified Ki67 and CD8 IHC in the same tumor regions. These new data demonstrate that there is a spatial correlation between Ki67 and CD8+ TILs in irradiated tumors and we include those data in the revised manuscript (see Supplementary Figure 6G).

Fig. 7I, J: If the distribution of CTLs and Tregs is different, it would be interesting to see the differences in intraepithelial and stromal distribution, respectively. It is important to discuss the effects of RT on the immune environment, which is currently the subject of many studies, in GEMMs.

Additional tumor areas were analyzed in order to have sufficient CTLs in separate epithelial and stromal compartments for robust statistical analysis. The new results (Supplemental Figure 7G in the revised manuscript) show increased CTLs in treated tumor relative to controls in both stromal and intraepithelial/tumoral compartments, though the increase was more obvious in the intraepithelial/tumoral compartment. Tregs were only observed in the stromal compartment, so we did not do a new analysis for Tregs by compartment. The quantitation shown in Figure 7J of the revised manuscript represents Tregs in the stromal compartment only.

Line 437: As shown in Figure 6H, since there was no survival benefit in the early treatment model (study 2), this part seems to be somewhat overstated.

In both prostate and rectal cancer models a survival benefit was observed. While survival was prolonged only modestly in the aggressive *Pten;Trp53^{pc-/-}* model, study 1 was well powered, and in our view convincingly shows a survival benefit with radiation. We should point out that the number of animals used in study 2 was based on the assumption that we would see a greater survival benefit than in study 1, which was not what we found. In the end study 2 was likely underpowered to show a survival benefit. If we combine the results of study 1 and 2, the survival benefit holds for the 7.5 Gy x 5 fraction group overall. We do not think this analysis alters the conclusions of the study. However, we can add the graph below to the revised manuscript if the Reviewer or Editor disagree.

Line 457: Even if small autochthonous tumors have radioresistant persister cells, it wouldn't be able to explain the reason why the effect of radiotherapy seen in larger tumors is diminished in small tumors. Instead, is there a biological basis for the radioresistance particular to small tumors? Or perhaps it only reflects a rise in deaths brought on by side effects? Please discuss these issues.

We did not observe deaths due to side effects of radiation in study 2. The only apparent toxicity in tumor bearing mice was penile prolapse (2 of 6 animals). At necropsy all animals showed evidence of urinary tract obstruction (distended bladder, hydronephrosis, hydroureter). As discussed above, one possible explanation for why SABR-treated animals in study 2 appeared to benefit less from treatment than animals in study 1 is that animals were treated earlier, and thus radioresistant persister cells had more time to repopulate the tumor resulting in slightly earlier obstructive symptoms. This fits with the trend toward larger tumor size in animals treated at a younger age. Another possibility is that smaller tumors are less likely to respond to radiation due to underlying biological differences (e.g. proliferation rate). We now discuss these interesting possibilities in the revised manuscript (page 21-22). We have also considered the possibility that tumors could be arising de novo after the radiation, given that Cre is constitutively expressed; however, multiple groups have shown in this model that Cre becomes expressed between 3-6 weeks of age and therefore we do not think this is a likely explanation.

Minor:

Please unify the notation of Ki67 and Ki-67 in entire the manuscript.

We apologize for the error. Ki67 and Ki-67 are used interchangeably throughout the literature and there does not appear to be a consensus, and in the revised manuscript we unify the notation as Ki67.

Line 421: Is it "one" instead of "on"?

Thank you. This has been corrected in the revised manuscript.

Line 532: The axillary LNs seem to show a significant increase in Th in both males and females in the irradiated group. This may not be consistent with the description in the Discussion.

Agreed. We have edited the Discussion to more accurately reflect these data (page 26). The primary conclusions are unchanged, namely that lymphocytes in lymph nodes appear less radiosensitive than lymphocytes in circulation.

Fig. S3D: The explanation seems inadequate. The organs and doses for each image could be clearly indicated in the figure or legends.

Thank you for pointing this out. The headers for this figure as well as Figure S3C were inadvertently deleted during reformatting prior to submission of the original manuscript. This has been corrected in the revised manuscript.

Fig S3H: Does red dots show 7.5 x 5? Furthermore, the entire manuscript should clarify which RT schedule is for 7.5 Gy x 5 fr.

This was also an error in figure formatting that has been corrected in the revised manuscript. Except for the single comparator group shown in Table 1, all 5 fraction regimens were delivered QOD. To avoid confusion, this notation has been added throughout the manuscript either in the figure, figure legend, or text where appropriate.

Fig. S3J: It would be better to show images of the pelvis and lower abdomen separately, as they appear to have different irradiated fields.

This figure is referring only to mice treated to the pelvic field. This has been clarified in the figure legend in the revised manuscript (now Supplementary Figure 3I).

Line 192: The expressions of "DL4" in the manuscript are confusing. Please describe the dose and the fractions.

We recognize that it was confusing to define the dose levels in a Supplementary Table, and as noted above this information is now in Table 1 in the main manuscript. We have also worked to improve clarity throughout the text whenever we refer to dose levels. When only a single fractionation regimen is used, the dose and fractions are written out rather than referring to the dose level.

Fig S4C: Next to images, it is easier to understand if it is also represented graphically.

A graph of lymph node cross-sectional size has been added to the figure as suggested.

Fig 5A: It would be helpful to be described the irradiation dose.

The dose and regimen has been added to the figure legend of the revised manuscript.

Fig 6: The order is back and forth in the manuscript, so correct it.

We apologize for any confusion. The figure panels are presented in sequential order; however, it is sometimes necessary to refer to an earlier panel when describing the data. Hopefully the presentation is clearer in the revised manuscript.

Response to Reviewer #2:

This is an interesting large scale study on the use of hypofractionation on genetically modified mice. I find the results and conclusions very interesting and useful to the field. I also think it was an excellent choice to deliver these doses without anesthesia, as using anesthesia can influence the radiation response. The dosimetry has also been well worked out. I was hoping for a couple of points of clarification regarding the scope of the study. In addition, I have some minor suggestions for improvement.

We thank the Reviewer for the time they spent carefully considering our work and are happy that they find the results interesting and useful. The comments from the Reviewer were very helpful in considering how to improve presentation of the data as outlined in response to each point raised below.

The largest criticism of this paper that I have is the premise that this is SABR. In my mind, SABR is not only dose escalation, but also high conformity. All treatments described in this study are done with a Cs-137 source in a simple AP-PA beam arrangement. In my mind, this is a hypofractionation study, not SABR. That being said there are still a great deal of important findings in this paper. Understanding both the treatment response to the target area and the dose tolerances to OARs is vital information for using this technique on genetically modified mice. I think that reframing this as a study on hypofractionation would be more representative of the study described here. It should also be noted here that Cs-137 sources are becoming less popular in the field, as small animal irradiators based on x-ray tubes are being phased in. Expanding this study using a small animal irradiator with onboard imaging would greatly improve the calculated dose statistics that are used in this study.

We appreciate the Reviewer's careful reading and thoughtful comments about how best to present our findings, and we agree that our platform is not completely identical to SABR in clinical practice. We comment separately on the Reviewers specific points about distinctions between our data and clinical practice below:

1) *SABR is not only dose escalation, but also high conformity*: The platform we developed is able to achieve accurate and precise localization of high doses of radiation in the mouse. While our platform is not designed for highly conformal treatment of tumors, we would point out that all available mouse irradiation systems have conformality limitations. Moreover, with highly conformal systems, there is a risk of sacrificing tumor control as we find that even with high-resolution imaging it is challenging to define the borders of autochthonous tumors in mice. In fact, our experience with a variety of autochthonous tumors is that they are infiltrative and extend beyond what is visible on MRI or other imaging modalities, making it necessary to treat some normal tissue through which the tumor invades in order to target all local disease. Therefore, we think that the approach to understand the effects of high dose radiation on both tumor and normal tissue using localized, but not entirely conformal treatment, is of value. Any clinical SABR treatment also has practical limits to achievable conformality and invasive tumor treatments will purposefully include normal tissue in the field. Thus, the data on the toxicity of SABR treatments and tolerance of normal tissues is important to consider when treating autochthonous tumors and is something we now point out in the Discussion of the revised manuscript (page 23).

2) *Expanding this study using a small animal irradiator with onboard imaging would greatly improve the calculated dose statistics that are used in this study*: While irradiations that are possible with kV-energy, image-guided machines are potentially more conformal, there are also intrinsic limitations to these machines. i) cost and time required to conduct a study such as the one we present would be prohibitive for many investigators. ii) kV irradiation of the mouse can produce inhomogeneous dose distribution and kV irradiation has high surface skin dose with high photoelectric effect impacting bone dose and the hematopoietic system. For these reasons, kV irradiators, although potentially more conformal, are still not a perfect representation of clinical SABR. This point is now discussed in the revised manuscript (page 23).

For the sake of simplicity, we initially chose to use the term SABR throughout the manuscript. Nevertheless, the Reviewer raises an important point, and to avoid any misconceptions with how the dose distribution and conformality of our preclinical radiotherapy platform compares to clinical delivery of SABR, we no longer use this term in the revised manuscript. Instead, we now refer to our treatment as stereotactic ablative radiotherapy (SART), which we believe is a more accurate description of our platform for the reasons discussed above.

Minor Points:

1) *In terms of the dose calculation presented in this paper, how was the CT density table established?*

Hounsfield units for the microCT scanner were calibrated annually to air, water, and bone-like material using a phantom provided by the manufacturer. MicroCT Hounsfield Units were used to classify the tissue type in each voxel, and to scale the physical density in each voxel based on a generic microCT density curve for the purposes of

Monte Carlo dose calculation. This description has now been added to the Methods of the revised manuscript (page 34).

2) This manuscript would greatly benefit from a table summarizing the dose regimens tested, and the associated tumor control / normal tissue toxicity. Please add this to the main manuscript.

We thank the Reviewer for this suggestion and have moved the relevant Supplementary Table to the main manuscript (formerly Supplementary Table 1, now Table 1 in the revised manuscript). The table has also been updated to include information regarding both incidence and timing of severe toxicity/death occurring after radiation. We agree this will help readers compare toxicity of the various dose regimens tested. A challenge in presenting more subtle toxicity is that there is no defined grading system for toxicity in mice. We are currently working on further characterizing radiation- and chemotherapy-induced toxicity by organ system in order to establish a grading system as a topic for future work. We did not add tumor control probability to the table because that is reported later in the manuscript, and in our view would be confusing to add to this table. We hope that readers will be able to easily glean this information from the KM plots shown in Figures 6 and 7.

3) Ki67 expression is featured in this paper but not explained. Can you please provide context for why this is important?

We apologize for not being clear. Ki67 is an immunohistochemical marker of proliferating cells. This is now better described in the revised manuscript (page 17).

4) Please provide more details on how the film dosimetry was conducted. Absolute film dosimetry can be tricky if not done very carefully. Typically new calibration curves are recommended with each new batch of film due to inter-batch variability.

The film dosimetry was performed using Gafchromic film. A calibration curve was derived by exposing pieces from the same batch of film to a full range of doses. The same film scanner was used for calibration and measurement, and the orientation of the film and time between exposure and scanning were kept consistent between calibration and measurement. The “Radiation delivery and dosimetry” section of the Methods has been updated to include this information (page 30). Of note, film dosimetry was conducted independently by 3 medical physicists over a period spanning 6 years, the results of which were congruent.

REVIEWERS' COMMENTS:

Reviewer #1 (Remarks to the Author):

Minor points:

Fig 6M, S6F; These results are interesting. While a significant increase in CD8+ T cells overall was observed (Fig 6M), there was no significant difference in CD8+ T cells restricted to the periphery (Fig S6F). Does this suggest that the SART-induced increase in CD8+ T cells is mainly observed in the core of the tumor rather than in the periphery? If so, since the upregulation of CD8+ T cells' infiltration to the core, where originally CD8+ T cell infiltration was less, is informative for understanding the immune response to radiotherapy, please consider adding this possibility to the discussion section.

In the results section of Page 16, the reason for setting up study 2 was "To determine whether treating tumors at an earlier stage would improve outcomes". As a result, the early intervention did not contribute to prolonged survival. Therefore, it would be helpful to add the survival curves of the Rebuttal letter to the manuscript to support the author's argument more directly in the discussion section.

In the revised version, the authors seem to have used "SART" instead of "SABR" to refer to the method used in this study, and to use "SABR" only to mention clinical settings. This change would help readers to avoid confusion.

Please clarify the abbreviation of SBRT as "stereotactic body radiation therapy".

Please add Fig S7G to the main manuscript.

Reviewer #2 (Remarks to the Author):

Thank you for addressing the concerns that I outlined on my previous review. For the most part my criticisms have been addressed. Given the capabilities of your equipment, the fact that the animals are immobilized, and that image guidance was used in the original plan, I would say the current framing of the manuscript is appropriate.

As an aside, the acronym SABR stands for Stereotactic Ablative Radiotherapy. Changing the acronym to SART and applying it to the same term does not change this aspect of the manuscript.

My concerns have been adequately addressed. I have no objection to publishing this in its current form.

Response to reviewers (point by point response to reviewers in blue).

Reviewer #1 (Remarks to the Author):

Minor points:

Fig 6M, S6F; These results are interesting. While a significant increase in CD8+ T cells overall was observed (Fig 6M), there was no significant difference in CD8+ T cells restricted to the periphery (Fig S6F). Does this suggest that the SART-induced increase in CD8+ T cells is mainly observed in the core of the tumor rather than in the periphery? If so, since the upregulation of CD8+ T cells' infiltration to the core, where originally CD8+ T cell infiltration was less, is informative for understanding the immune response to radiotherapy, please consider adding this possibility to the discussion section.

Indeed, the data in the prostate cancer model suggest that radiation may allow Tc cells to penetrate deeper into the tumor. We thank the reviewer for highlighting this and we have added it to the discussion section as we finalized the manuscript.

In the results section of Page 16, the reason for setting up study 2 was "To determine whether treating tumors at an earlier stage would improve outcomes". As a result, the early intervention did not contribute to prolonged survival. Therefore, it would be helpful to add the survival curves of the Rebuttal letter to the manuscript to support the author's argument more directly in the discussion section.

The survival curve presented in the rebuttal letter was a compilation of the survival curves shown in Fig 6F and 6H, which were presented specifically in response to the Reviewer's concern regarding "overstatement" of radiation showing survival benefit in autochthonous models (line 437 of the original submission). The purpose was to point out that there is a small, but significant survival benefit in the prostate cancer model considered even when combining results of the early and late intervention. Since a survival benefit is already demonstrated by study 1 (late intervention, Fig 6F) we do not think adding the combined survival curve (study 1 sham and 7.5Gy x5 arms + study 2) changes the conclusion. Furthermore, since these were separate studies, we think it is best to present the survival curves separately as is shown in Fig 6F and 6H.

In the revised version, the authors seem to have used "SART" instead of "SABR" to refer to the method used in this study, and to use "SABR" only to mention clinical settings. This change would help readers to avoid confusion.

Please clarify the abbreviation of SBRT as "stereotactic body radiation therapy".

We thank the reviewer for pointing this out. The acronym SBRT is now defined in the introduction paragraph.

Please add Fig S7G to the main manuscript.

These panels have now been added to the main manuscript as requested.

Reviewer #2 (Remarks to the Author):

Thank you for addressing the concerns that I outlined on my previous review. For the most part my criticisms have been addressed. Given the capabilities of your equipment, the fact that the animals are immobilized, and that image guidance was used in the original plan, I would say the current framing of the manuscript is appropriate.

As an aside, the acronym SABR stands for Stereotactic Ablative Radiotherapy. Changing the acronym to SART and applying it to the same term does not change this aspect of the manuscript.

My concerns have been adequately addressed. I have no objection to publishing this in its current form.

We thank the Reviewer again for their thoughtful review and insightful comments that have helped to improve data presentation.